# Chronic exposure to odors at naturally occurring concentrations triggers limited plasticity in early stages of *Drosophila* olfactory processing

Zhannetta V Gugel, Elizabeth G Maurais, Elizabeth J Hong*

Division of Biology and Biological Engineering, California Institute of Technology, Pasadena, United States

*For correspondence:
ejhong@caltech.edu

**Abstract** In insects and mammals, olfactory experience in early life alters olfactory behavior and function in later life. In the vinegar fly *Drosophila*, flies chronically exposed to a high concentration of a monomolecular odor exhibit reduced behavioral aversion to the familiar odor when it is reencountered. This change in olfactory behavior has been attributed to selective decreases in the sensitivity of second-order olfactory projection neurons (PNs) in the antennal lobe that respond to the overrepresented odor. However, since odorant compounds do not occur at similarly high concentrations in natural sources, the role of odor experience-dependent plasticity in natural environments is unclear. Here, we investigated olfactory plasticity in the antennal lobe of flies chronically exposed to odors at concentrations that are typically encountered in natural odor sources. These stimuli were chosen to each strongly and selectively excite a single class of primary olfactory receptor neuron (ORN), thus facilitating a rigorous assessment of the selectivity of olfactory plasticity for PNs directly excited by overrepresented stimuli. Unexpectedly, we found that chronic exposure to three such odors did not result in decreased PN sensitivity but rather mildly increased responses to weak stimuli in most PN types. Odor-evoked PN activity in response to stronger stimuli was mostly unaffected by odor experience. When present, plasticity was observed broadly in multiple PN types and thus was not selective for PNs receiving direct input from the chronically active ORNs. We further investigated the DL5 olfactory coding channel and found that chronic odor-mediated excitation of its input ORNs did not affect PN intrinsic properties, local inhibitory innervation, ORN responses or ORN-PN synaptic strength; however, broad-acting lateral excitation evoked by some odors was increased. These results show that PN odor coding is only mildly affected by strong persistent activation of a single olfactory input, highlighting the stability of early stages of insect olfactory processing to significant perturbations in the sensory environment.

## Editor's evaluation

Does exposure to odors induce some type of adaptive plasticity at the early layers of the olfactory circuit? In this valuable work, the authors use naturalistic concentrations of odor to provide convincing evidence that there is actually very little plasticity in the projection neurons at the glomerular layer of the system. Their findings show small increases in responses to low concentrations of the exposed odor.

## Introduction

In many animals, early sensory experience modifies the structure and function of sensory systems. For example, visual experience is required for the normal development of the mammalian visual system (*Espinosa and Stryker, 2012*). One prominent hypothesis is that an important function of sensory plasticity is to adapt circuit function to the current statistical distribution of sensory inputs in the environment, allowing for more efficient sensory codes (*Barlow, 1961*; *Fiser et al., 2010*; *Gilbert et al., 2009*; *Pienkowski and Eggermont, 2011*). This hypothesis requires sensory driven plasticity to be stimulus- and cell-specific; in other words, neurons encoding specific stimuli that occur very frequently (or very rarely) in the environment should be selectively affected by plasticity (*Das et al., 2011*; *Kreile et al., 2011*; *Sachse et al., 2007*; *Sengpiel et al., 1999*; *Wilson et al., 1985*; *Zhang et al., 2001*). The circuit mechanisms that would implement such use-dependent, input-specific plasticity are not well understood.

The orderly structure of the olfactory system provides a useful experimental model for investigating the synaptic and circuit mechanisms mediating stimulus-selective sensory plasticity. In insect and vertebrate olfactory circuits, sensory information is organized in anatomically discrete synaptic compartments, called glomeruli. Each glomerulus receives direct input from only a single class of primary olfactory receptor neurons (ORNs), all expressing the same olfactory receptor, and, thus, all sensitive to the same chemical feature(s) (*Ressler et al., 1994*; *Vassar et al., 1994*; *Vosshall and Stocker, 2007*; *Vosshall et al., 2000*). Furthermore, the dendrites of each second-order uniglomerular projection neuron (PN) arborize in only a single glomerulus, so each PN receives direct input from only a single class of ORNs (*Stocker et al., 1990*). In the vinegar fly *Drosophila melanogaster*, the majority of odorant receptors and ORN subtypes have been mapped to their corresponding glomeruli in the brain (*Couto et al., 2005*; *Fishilevich and Vosshall, 2005*; *Silbering et al., 2011*), and the odor tuning profiles for a large subset of the odorant receptors have been characterized (*Silbering et al., 2011*; *de Bruyne et al., 1999*; *de Bruyne et al., 2001*; *Hallem and Carlson, 2006*; *Hallem et al., 2004*). As a result, specific odors can be used to selectively target neural activation of defined olfactory channels (*Olsen et al., 2010*; *Schlief and Wilson, 2007*). Together with the highly compartmentalized organization of the circuit, these features make the fly olfactory system a powerful experimental model for studying the specificity of sensory plasticity.

Passive odor experience in early life, in the absence of explicit coupling to reward or punishment, can alter olfactory circuit structure and function, including olfactory preference or discrimination ability (*Mandairon and Linster, 2009*; *Mandairon et al., 2006a*). For instance, chronic odor exposure in rodents can trigger changes in the structural connectivity and physiological response properties of neurons in the olfactory bulb, the first central processing area for odors in the brain (*Wilson et al., 1985*; *Liu and Urban, 2017*; *Liu et al., 2016*; *Todrank et al., 2011*; *Woo et al., 2006*). In insects, passive odor experience has also been found to impact olfactory processing in the antennal lobe, the insect analog of the olfactory bulb (*Golovin and Broadie, 2016*; *Twick et al., 2014*; *Bai and Suzuki, 2020*). Chronic exposure to high concentrations of monomolecular odor, stimuli which are typically strongly aversive, reduces behavioral avoidance toward the familiar odor selectively when it is reencountered (*Das et al., 2011*; *Devaud et al., 2001*; *Devaud et al., 2003*); in parallel, structural alterations in glomerular volume are also observed (*Das et al., 2011*; *Sachse et al., 2007*; *Devaud et al., 2001*; *Devaud et al., 2003*; *Golovin et al., 2019*; *Kidd et al., 2015*). Odor experience-dependent reductions in olfactory aversion have been interpreted as a form of long-term behavioral habituation to overrepresented stimuli in the environment and are attributed to reduced PN responses in flies chronically exposed to odor (*Das et al., 2011*; *Sachse et al., 2007*; see also *Kidd et al., 2015*). Importantly, since behavioral plasticity is odor-specific and does not generalize to other odors, structural and functional neural plasticity are also believed to be glomerulus-specific (and thus odor-specific), acting to selectively reduce the olfactory sensitivity of only those PNs activated by the overrepresented stimuli. However, since past studies exposed animals to intense monomolecular odors (*Das et al., 2011*; *Sachse et al., 2007*; *Devaud et al., 2001*; *Devaud et al., 2003*; *Golovin et al., 2019*; *Chodankar et al., 2020*; e.g. from odor sources of 10% isoamyl acetate [v/v] or 20% ethyl butyrate [v/v]), which typically excite many classes of ORN inputs, stringent testing of this idea has not been possible.

Arriving at a systematic understanding of how odor experience modifies the structure and function of olfactory circuits has been challenging for several reasons. First, diverse protocols are used for

odor exposure, which vary in the degree of control over odor delivery, odor concentration, timing, as well as context (availability of food, mates, etc). Second, different studies focus on different odors and glomeruli, and the high dimensionality of chemical stimulus space and olfactory circuits presents unique challenges to methodical exploration. Finally, nearly all studies, in insects or in mammals, use very high concentrations of monomolecular odorants during the exposure period, at intensities that are not encountered in the natural world (see Discussion). Odors at these concentrations broadly activate many classes of ORNs, complicating the evaluation of the contributions of direct and indirect activity for triggering plasticity in each olfactory processing channel. Some of the major outstanding questions include: (1) How does olfactory plasticity modify circuit function in the context of odor environments that could be realistically encountered in the natural world? (2) To what extent is plasticity selective for the olfactory channel(s) which directly detect overrepresented odors? (3) Are the rules governing olfactory plasticity the same or different across glomeruli?

The goals of this study were to investigate the impact of olfactory experience on odor coding in the *Drosophila* antennal lobe using a physiologically plausible olfactory environment that strongly but selectively increases neural activity in a single class of ORNs. This experimental design allowed us to readily distinguish the effects of direct versus indirect chronic activity on specific classes of PNs, which convey neural output from the antennal lobe to higher olfactory centers in the fly brain. This distinction is important because it allows us to unambiguously evaluate whether olfactory plasticity selectively affects only the chronically active glomerulus. Investigating three different glomerular channels, we unexpectedly observed that, rather than reduce PN sensitivity, chronic odor exposure resulted in mild increases in PN sensitivity, particularly in response to weak odors. PN responses to moderate or strong stimuli were mostly unaffected. Changes in odor responses, when present, were observed broadly, both in glomeruli that receive either direct or indirect activity from the chronically active ORN class. These results diverge from current models suggesting that odor-specific behavioral plasticity arises from a glomerulus-specific long-term adaptation of PN responses in the antennal lobe (*Das et al., 2011*; *Sachse et al., 2007*; *Chodankar et al., 2020*; *Sudhakaran et al., 2012*) and motivate the search for other circuit mechanisms mediating how olfactory experience alters behavior toward familiar odors.

## Results

### Chronic activation of direct ORN input modestly increases PN responses to weak stimuli

To investigate how odors that are overrepresented in the environment are encoded by the fly olfactory system, we chronically exposed flies to 1 s pulses of a specific monomolecular odorant, introduced into the bottle in which they normally grow (*Figure 1A*). We chose to use the odors at concentrations previously shown to selectively activate a single class of ORNs (*Olsen et al., 2010*; *Hong and Wilson, 2015*) in order to simplify the investigation of odor coding in postsynaptic PNs receiving direct versus indirect chronic excitation. An additional criterion was that the odor stimuli drive strong, consistent, and saturating levels of firing in PNs receiving direct input from the activated ORN class (in other words, using a higher concentration of odor would not reliably drive higher firing rates in the PN type that is directly excited by that stimulus). Photoionization measurements of the odor concentration in the rearing bottle demonstrated that the stimulus amplitude changed <20% over 12 hr (*Figure 1B*), after which the odor source was refreshed. The largest decrease in delivered odor concentration occurred within the first 2 hr and slowed significantly thereafter. Odors were pulsed to minimize neural adaptation to the odor, and the interval between odor pulses delivered to the bottle was 20 s. Pilot recordings showed that, at this interstimulus interval, the odors reliably activated cognate PNs to saturating or near saturating firing rates over many trials (*Figure 1C–D*, and data not shown; see also *Figure 2Q*). In addition, PN response duration to the 1 s stimulus increased modestly with successive odor presentations. Thus, although the odor stimuli to which we exposed flies were of significantly lower concentration than what has been previously used to investigate the effect of chronic odor exposure on PNs (*Das et al., 2011*; *Sachse et al., 2007*; *Kidd et al., 2015*), these stimuli drove high, reliable, near saturating levels of firing in their corresponding PN type (*Figure 1C*, data not shown). On average, this odor exposure protocol more than doubled the average firing rate of the targeted

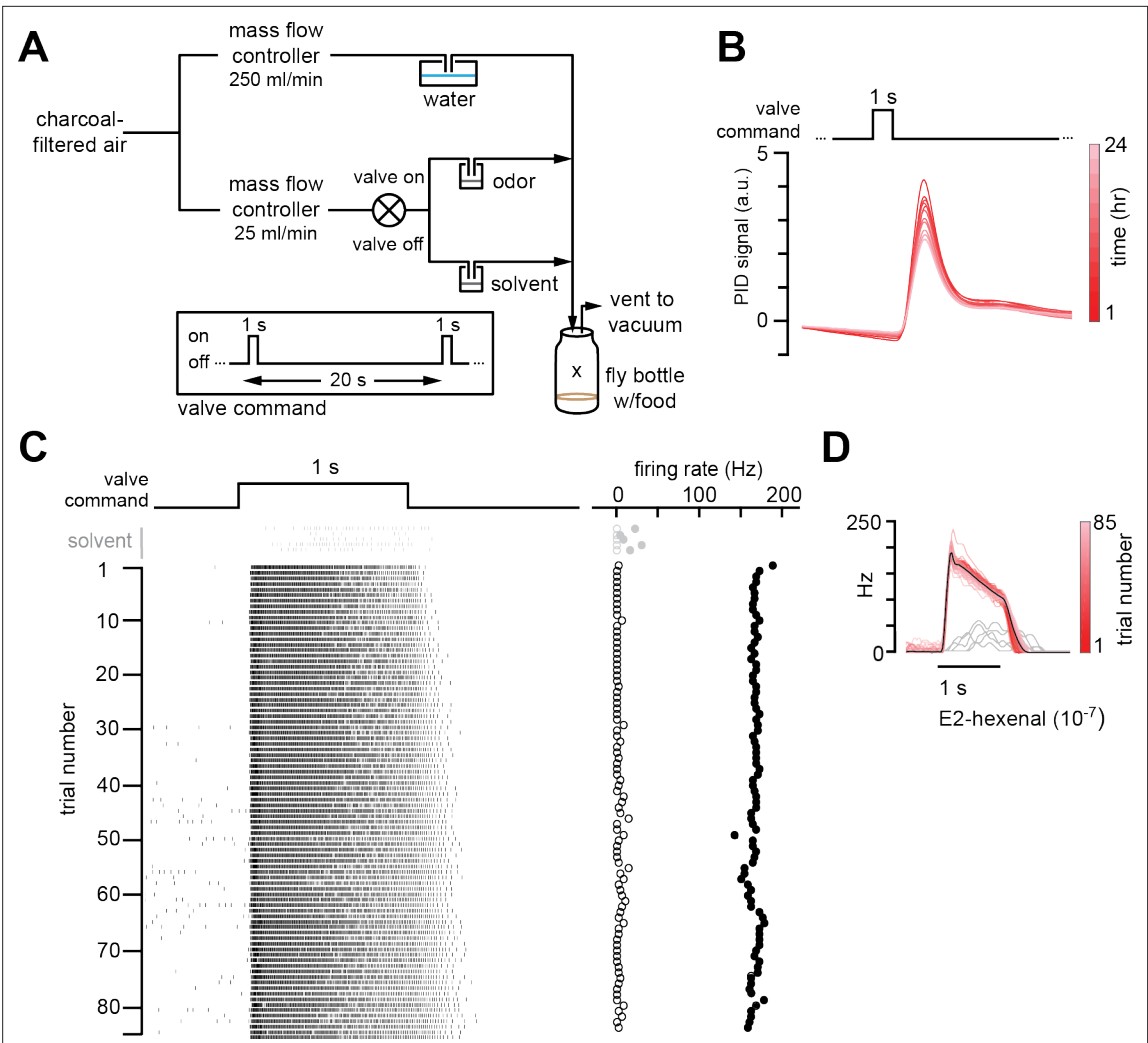

**Figure 1.** Chronic stimulation of olfactory neurons in a controlled odor environment. (**A**) Schematic of experimental setup for chronic odor exposure. The valve was opened for 1 s every 20 s to deliver odor. See Materials and methods (*chronic odor exposure*) for details. (**B**) Photoionization detector measurement of odor concentration at the center of the fly bottle ("X" in **A**) during chronic odor exposure to 2-butanone ($10^{-4}$) over the course of 24 hr. Each trace is the odor profile averaged across 45 consecutive odor pulses (collected over 15 min), sampled every 2 hr over a 24 hr period. (**C**) Raster plots of the spiking responses in an example DL5 projection neuron (PN) to a 1 s pulse of either solvent (paraffin oil, gray) or E2-hexenal ($10^{-7}$, black) in consecutive trials spaced 20 s apart. Recordings were established from a fly immediately after 2 days of chronic exposure to E2-hexenal ($10^{-7}$) as in (**A**). Spontaneous (open circles) and stimulus-evoked (filled circles) firing rates are plotted for each trial. (**D**) Peristimulus time histograms of measurements in (**C**) show that the odor environment reliably evokes high levels of PN firing across presentation trials. The average odor-evoked response across all trials is in black. Responses to presentation of solvent are overlaid in gray.

PN and elicited more than a million extra spikes in the activated PN over the course of the 2-day period of exposure (see Materials and methods).

Using these conditions, newly eclosed flies were chronically exposed for 2 days to E2-hexenal ($10^{-7}$), which selectively activates ORNs projecting to glomerulus DL5, or to solvent (as a control) (see Materials and methods). On day 3, we established fluorescently guided, whole-cell current clamp recordings from uniglomerular PNs receiving direct input from the DL5 glomerulus (hereafter referred to as DL5 PNs, *Figure 2A*), and measured their responses to a concentration series of E2-hexenal. DL5 PNs were identified and targeted for recording based on their expression of green fluorescent protein (GFP), mediated by a genetic driver that specifically labels this cell type (see Materials and methods). DL5 PN responses elicited by moderate to high concentrations of E2-hexenal (>$10^{-8}$) were unchanged in E2-hexenal exposed flies, as compared to controls (*Figure 2B and Q*). However, low concentrations of E2-hexenal ($10^{-10}$–$10^{-9}$) elicited increased levels of odor-evoked membrane depolarization in DL5

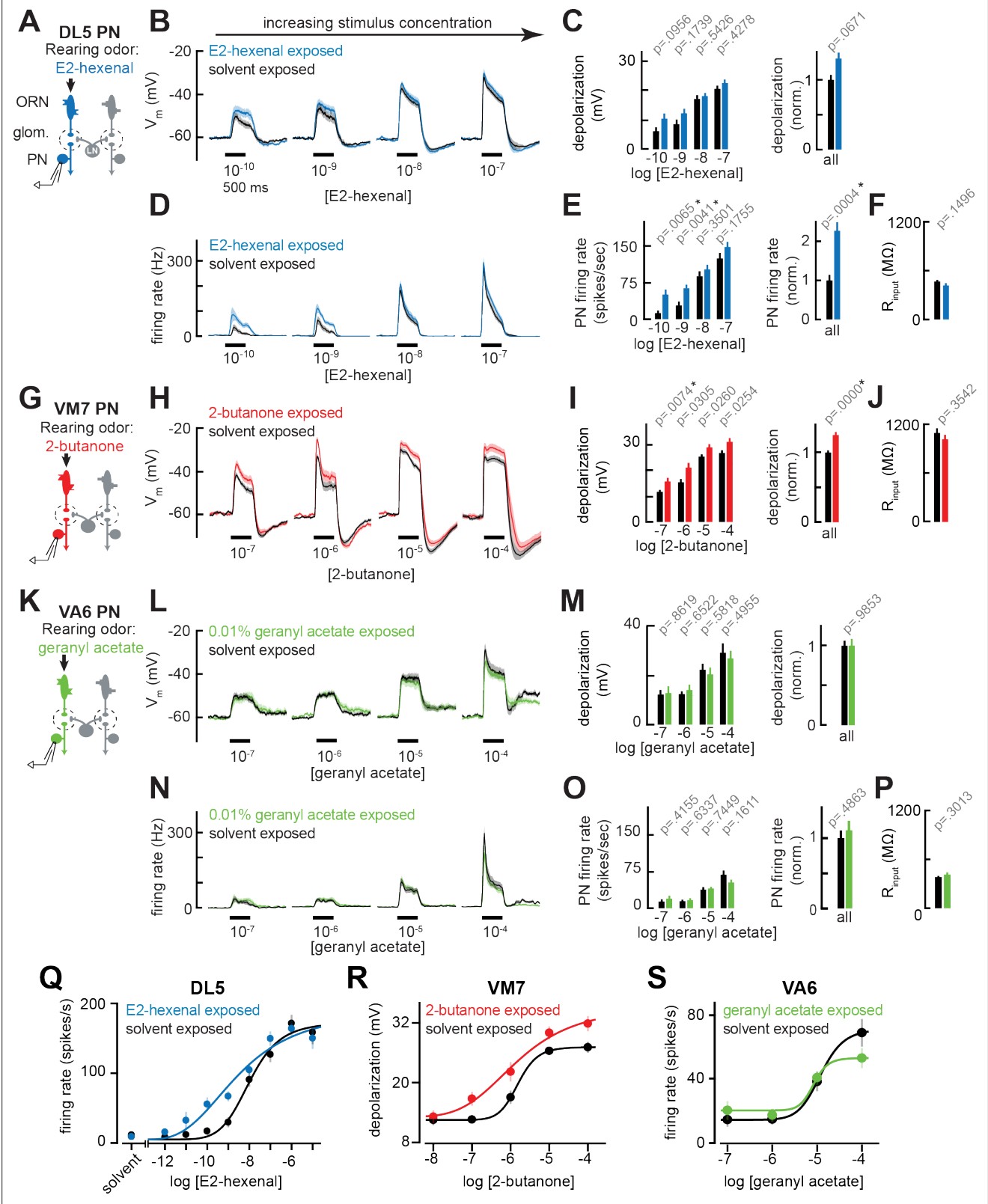

**Figure 2.** Chronic excitation of direct presynaptic olfactory receptor neuron (ORN) input can modestly enhance projection neuron (PN) sensitivity to weak stimuli. (**A**) Schematic of experimental setup for (**B–F**). Recordings were established from DL5 PNs that receive direct presynaptic input from the ORN class (ab4a) chronically activated by the rearing odor (E2-hexenal, 10⁻⁷), n=7–19 cells. (**B**) Odor-evoked depolarization in DL5 PNs in response to varying concentrations of E2-hexenal from flies chronically exposed to E2-hexenal or solvent (paraffin oil). (**C**) *Left:* mean odor-evoked depolarization

*Figure 2 continued on next page*

*Figure 2 continued*

to each stimulus in (**B**) in the 500 ms after nominal stimulus onset. *Right:* mean normalized odor-evoked depolarization across all stimuli. Within each stimulus, responses were normalized to the mean odor-evoked depolarization in the solvent-exposed group. (**D**) Peristimulus time histograms of odor-evoked spiking responses in DL5 PNs from (**B**). (**E**) *Left:* mean odor-evoked firing rates to each stimulus from (**D**) in the 500 ms after nominal stimulus onset. *Right:* mean normalized odor-evoked firing rates across all stimuli, computed analogously as in (**C**). (**F**) Mean input resistance of DL5 PNs recorded from E2-hexenal- or solvent-exposed flies. (**G**) Experimental setup for (**H–J**). Recordings were established from VM7 PNs that receive direct presynaptic input from the ORN class (pb1a) chronically activated by the rearing odor (2-butanone, $10^{-4}$), n=6–11 cells. (**H–I**) As in (**B–C**) but describing odor-evoked depolarization in VM7 PNs in response to a concentration series of 2-butanone from flies chronically exposed to 2-butanone or solvent. (**J**) Mean input resistance of VM7 PNs recorded from 2-butanone- or solvent-exposed flies. (**K**) Experimental setup for (**L–P**). Recordings were established from VA6 PNs that receive direct presynaptic input from the ORN class (ab5a) chronically activated by the rearing odor (geranyl acetate, $10^{-4}$), n=7–11 cells. (**L–P**) As in (**B–F**) but describing odor-evoked responses to varying concentrations of geranyl acetate in VA6 PNs from flies chronically exposed to geranyl acetate or solvent. (**Q–S**) Response curves in DL5 (**Q**), VM7 (**R**), and VA6 (**S**) PNs from odor-exposed and control flies to a concentration series of the cognate odor for each glomerulus. Results from panels (**A–P**) are replotted here and extended with measurements at additional stimulus concentrations. n=3–19 cells. All plots are mean ± SEM across flies in each experimental condition. One cell was recorded per fly. Two-tailed p-values report the fraction of resampled absolute differences of means (between simulated experimental groups) which are greater than the absolute observed difference between the means of experimental groups (odor-exposed versus solvent-exposed; see Materials and methods). Starred (*) p-values are significant at the level of α=0.05, corrected for multiple comparisons (Bonferroni adjustment). See *Supplementary file 1* for the full genotype and number of cells in every condition.

The online version of this article includes the following source data for figure 2:

**Source data 1.** Source data for *Figure 2B–C, D–E, H–I, L–M and N–O*.

PNs from E2-hexenal exposed flies compared to solvent-exposed flies (*Figure 2B*). The heightened depolarization of DL5 PNs by weak stimuli corresponded to higher average rates of odor-evoked spiking (*Figure 2D*).

We quantified these effects by calculating the total depolarization and average evoked firing rate during the first 500 ms after nominal stimulus onset (*Figure 2C and E*). To determine if any differences were arising by chance, we used permutation testing in which we iteratively shuffled the experimental labels of each measurement (E2-hexenal versus solvent exposure) within each stimulus. p-Values were calculated directly from the fraction of 10,000 shuffled trials in which the absolute difference between the simulated group means was larger than or equal to the actual observed absolute mean difference (see Materials and methods). This statistical analysis confirmed that E2-hexenal exposure increased odor-evoked firing rates in DL5 PNs to weak but not strong levels of stimulation (*Figure 2E*). When firing rates in E2-hexenal exposed flies were normalized to the control rate within each stimulus, we observed an overall increase in odor-evoked DL5 PN firing rate due to E2-hexenal exposure (*Figure 2E-F*). Differences in the amount of membrane depolarization between odor- and solvent-exposed groups were not statistically significant at any stimulus concentration (*Figure 2C*), suggesting a small, but systematic increase in membrane depolarization was nonlinearly amplified by its interaction with the firing threshold in DL5 PNs.

We next asked whether these results generalize to other glomeruli. Using the same approach, we exposed flies to either 2-butanone ($10^{-4}$), which selectively activates ORNs projecting to glomerulus VM7 (*Figure 2G*), or geranyl acetate ($10^{-4}$), which selectively activates ORNs projecting to glomerulus VA6 (*Figure 2K*). The concentrations of these odors were chosen because each stimulus selectively and reliably elicits high firing rates (>100–150 Hz) in its corresponding PN, comparable to the level of E2-hexenal ($10^{-7}$) activation of DL5 PNs. Again, we chronically exposed flies (separate groups) for 2 days to each of these stimuli and measured the responses in each PN type (corresponding to the glomerulus receiving direct input from the activated ORNs) to a concentration series of each odor (*Figure 2G and K*). Like the DL5 glomerulus, VM7 PNs in 2-butanone-exposed flies exhibited modest increases in odor-evoked depolarization in response to direct stimulation (by 2-butanone) as compared to control flies, and these effects were more pronounced at weak concentrations ($10^{-7}$; *Figure 2H–I and R*). VM7 PN spikes are small and filtered in comparison to those of other PNs, and odor-evoked spikes riding on large depolarizations could not be reliably counted across all firing rates in our data set. Therefore, for VM7 PNs only, we report odor responses only in terms of membrane depolarization.

However, chronic activation of direct ORN input to VA6 PNs by exposure to geranyl acetate did not alter PN odor responses across the entire range of concentrations of the familiar odor geranyl acetate, neither at the level of membrane depolarization nor firing rate (*Figure 2L–P and S*). These concentrations elicited levels of membrane depolarization (ranging from 10 to 30 mV) which were

similar to that at which other PN types exhibited plasticity after chronic exposure. Together, these results demonstrate that, in some glomeruli like DL5 and VM7, chronic activation of direct ORN input modestly enhanced PN odor responses to weak stimuli, resulting in an overall expansion of the range of stimulus concentrations dynamically encoded by PN activity. However, this effect does not appear to be universal across all glomeruli.

## Chronic excitation of ORNs in other glomeruli alters PN response properties

Although olfactory input is compartmentalized into feedforward excitatory channels organized around each glomerular unit, odor processing depends on an extensive network of local neurons (LNs) that mediate lateral excitatory and inhibitory interactions among glomeruli (*Wilson, 2013*). Thus, PN odor responses reflect both direct input from its presynaptic ORN partners, and indirect input, arising from activity in other glomeruli and received via local lateral circuitry. Having observed that chronic activation of an ORN class that provides direct input to a PN can elicit some plasticity in that PN, we next asked whether this plasticity is selective for those PNs directly postsynaptic to the chronically active ORNs, or whether PNs that receive only indirect activity from the chronically active ORN subtype are similarly affected. To address this question, we evaluated odor responses in VM7 and VA6 PNs from flies chronically exposed to E2-hexenal ($10^{-7}$; *Figure 3A and I*), which provides direct excitation to the DL5 glomerulus.

Chronic indirect excitation evoked plasticity with varying properties in different PNs. In E2-hexenal exposed flies, non-DL5 PNs had mildly enhanced responses to weak stimuli (*Figure 3B and J*). This effect was small but consistently observed in both VM7 and VA6 PNs at the level of odor-evoked depolarization (*Figure 3C and K*). In VM7 PNs, the baseline spontaneous firing rate also trended higher in E2-hexenal exposed flies as compared to controls (*Figure 3G–H*). Finally, chronic indirect excitation impacted the post-stimulus response properties of some PNs. For example, odor-evoked depolarization in VM7 PNs from E2-hexenal-exposed flies had a more pronounced and prolonged after hyperpolarization as compared to controls (*Figure 3B and E*), or as compared to flies that experienced chronic direct excitation (2-butanone exposed; *Figures 2H and 3F*). This effect does not appear to generalize to all glomeruli. VA6 PNs, for instance, exhibit comparatively different post-stimulus response dynamics, characterized by an epoch of delayed excitation that persists beyond odor offset (*Figure 3J*). In recordings from VA6 PNs in E2-hexenal exposed flies, this post-stimulus excitation was enhanced across multiple odor concentrations, as compared to solvent-exposed controls (*Figure 3J–K*). Most of these differences, however, were within the range of subthreshold depolarizations, and so the overall impact of chronic indirect excitation on VA6 firing rates was mild (*Figure 3M–N*). Together, these experiments demonstrated that chronic, focal excitation of a single ORN class can lead to changes in PN odor response properties in multiple glomeruli, including in glomeruli not receiving direct synaptic input from the chronically activated ORN class. This result implicates local lateral circuitry in the antennal lobe in activity-dependent PN plasticity after chronic exposure to odor.

## Chronic exposure to intense monomolecular odor can affect PN odor responses similarly to chronic exposure at naturalistic concentrations

The divergence of our results so far from those of some past studies, which concluded that chronic odor exposure elicits selective, long-lasting reductions in PN sensitivity (*Das et al., 2011*; *Sachse et al., 2007*), was surprising. Besides differing in how PN odor responses were measured (electrophysiology versus calcium imaging in past work), the major difference in our study is the odor environment in which flies are reared. Whereas past studies placed the odor source, a vial of concentrated monomolecular odor (1–20%), into the growth environment of the fly, we introduced monomolecular odors at comparatively lower concentrations as intermittent pulses into the growth bottle. Our reasoning was that odors at these lower concentrations still activate their corresponding PN type very strongly but are being presented at a concentration that could reasonably occur in a natural source; thus, the experiment more closely models how an odor overrepresented in a natural environment might impact olfactory function. However, given our unexpected results, we next decided to examine how chronic, sustained exposure to intense monomolecular odor affects PN responses.

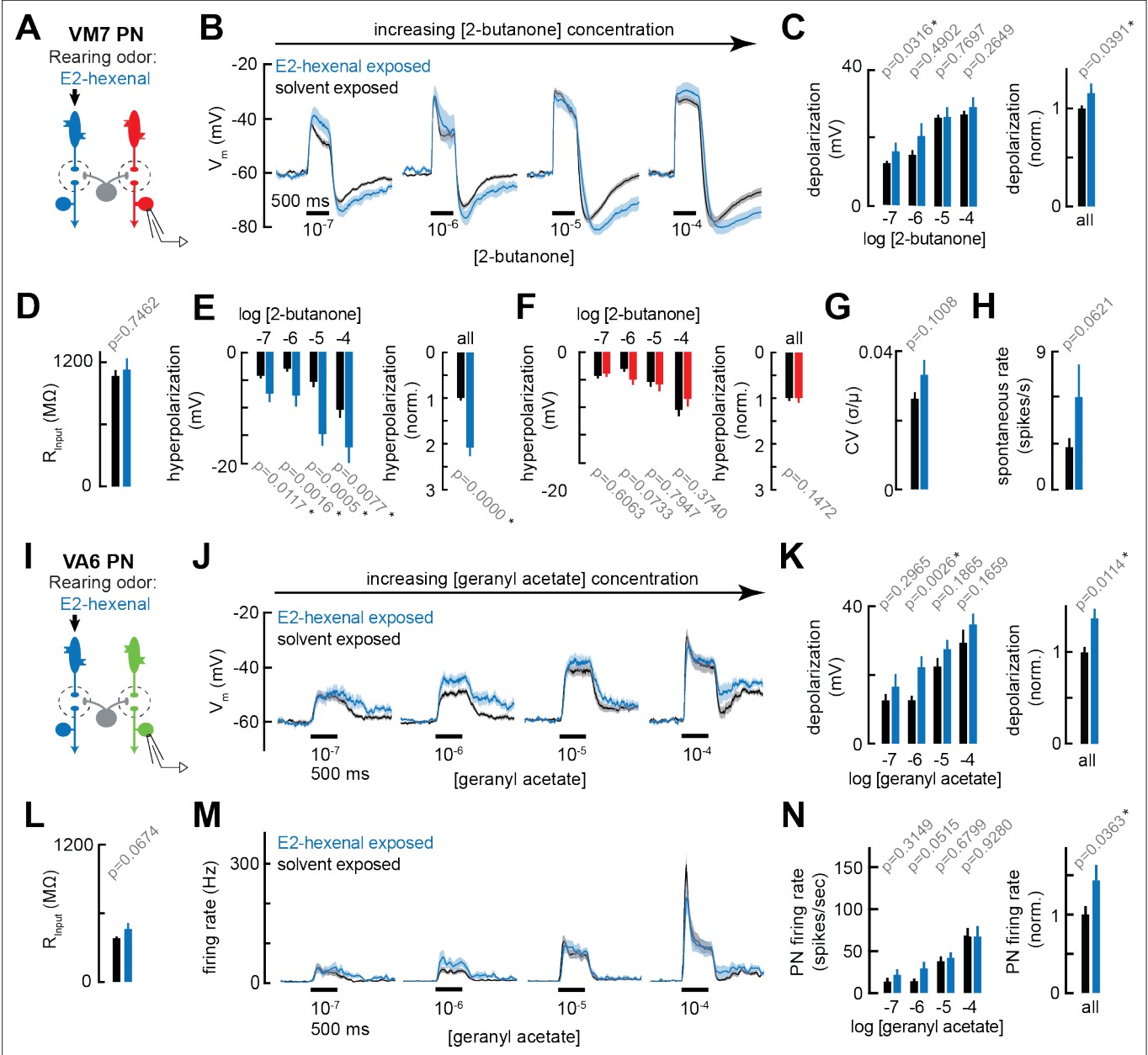

**Figure 3.** Chronic excitation of olfactory receptor neurons (ORNs) in other glomeruli alters projection neuron (PN) response properties. (**A**) Experimental setup for (**B–E**) and (**G–H**). Recordings were established from VM7 PNs that receive indirect activity from the ORN class (ab4a) chronically activated by the rearing odor (E2-hexenal, $10^{-7}$), n=3–11 cells. (**B**) Odor-evoked depolarization in VM7 PNs in response to varying concentrations of 2-butanone from flies chronically exposed to E2-hexenal or solvent (paraffin oil). (**C**) *Left:* mean odor-evoked depolarization to each stimulus from (**B**) in the 500 ms after nominal stimulus onset. *Right:* mean normalized odor-evoked depolarization across all stimuli. Within each stimulus, responses were normalized to the mean odor-evoked depolarization in the solvent-exposed group. (**D**) Mean input resistance of VM7 PNs recorded from E2-hexenal- or solvent-exposed flies. (**E**) *Left:* mean post-stimulus hyperpolarization in VM7 PN responses to each stimulus from (**B**), calculated over a 2.5 s window after stimulus offset. *Right:* mean normalized post-stimulus hyperpolarization across all stimuli. Within each stimulus, responses were normalized to the mean post-stimulus hyperpolarization in the solvent-exposed group. (**F**) Same as (**E**), but for VM7 PNs from **Figure 2H** with chronic activation of direct ORN input. Measurements are from VM7 PN recordings in flies chronically exposed to 2-butanone (red) or solvent (black), n=6–11 cells. (**G**) Mean coefficient of variation (CV) of membrane potential in VM7 PNs (from **B**), computed over the 5 s window before stimulus onset, in E2-hexenal- or solvent-exposed flies. (**H**) Mean spontaneous firing rate of VM7 PNs (from **B**) during the 5 s window before stimulus onset in E2-hexenal- or solvent-exposed flies. (**I**) Experimental setup for (**J–N**). Recordings were established from VA6 PNs that receive indirect activity from the ORN class (ab4a) chronically activated by the rearing odor (E2-hexenal, $10^{-7}$), n=5–11. (**J–K**) As in (**B–C**) but for odor-evoked depolarization in VA6 PNs to varying concentrations of geranyl acetate. (**L**) As in (**D**) but for VA6 PNs. (**M–N**) As in (**B–C**) but for odor-evoked spiking in VA6 PNs in response to varying concentrations of geranyl acetate

*Figure 3 continued on next page*

*Figure 3 continued*

(corresponding to J), from flies chronically exposed to E2-hexenal ($10^{-7}$) or solvent (paraffin oil). All plots are mean ± SEM across flies, one cell/fly, in each experimental condition. p-Values are as described in *Figure 2*. See *Supplementary file 1* for the full genotype and number of flies in every condition.

The online version of this article includes the following source data for figure 3:

**Source data 1.** Source data for *Figure 3B–C, E–F, J–K and M–N*.

We introduced into the growth bottle of the flies a small vial containing either 20% geranyl acetate (>1000-fold more concentrated than in *Figure 2K–P*) or solvent (control). The opening of the vial was sealed with fine, porous mesh to prevent flies from contacting the odor but allowing diffusion of the odor out of the vial. We chose to expose flies to 20% geranyl acetate because, even at high concentrations, this stimulus still preferentially excites VA6 ORNs, whereas other odors like E2-hexenal and 2-butanone strongly activate many different ORN classes as the concentration increases (*Hallem and Carlson, 2006*; *Schlief and Wilson, 2007*). Each group of flies was continuously exposed to 20% geranyl acetate or solvent for 4 days, as in prior studies, with the odor source replenished daily. Using the same approach as described above, we recorded from PNs directly postsynaptic to ORNs activated by geranyl acetate (VA6, *Figure 4A*) or PNs that receive indirect input from them (DM6, *Figure 4G*). We found that chronic, sustained exposure to 20% geranyl acetate had little or no effect on VA6 PN odor responses spanning the range of concentrations of geranyl acetate tested (*Figure 4B–F*). VA6 PN responses in 20% geranyl acetate-exposed flies consistently trended higher than in solvent-exposed flies, particularly at the level of odor-evoked membrane depolarization (*Figure 4B–C*), but these small differences were not statistically significant after correction for multiple comparisons. Odor responses in DM6 PNs, which are excited by valeric acid, were moderately increased in 20% geranyl acetate-exposed flies as compared to solvent-exposed control flies (*Figure 4H–L*). As was the case in our earlier experiments, these heightened responses were most pronounced for weaker odor stimuli (Figure H–K). These results indicate that uninterrupted, chronic exposure to an intense monomolecular odor, 20% geranyl acetate, for 4 days has only a moderate effect on PN odor coding, qualitatively similar to the impact of long-lasting but intermittent exposure to odors at lower, naturalistic concentrations (*Figures 2 and 3*). We did not observe reductions in PN olfactory sensitivity in flies reared in any of the four odorized environments we studied. Together, these results show that prior reports indicating that chronic excitation reduces PN sensitivity do not, at a minimum, hold true for all glomeruli. In all cases, we examined – DL5, VM7, and VA6 – the overall effect of chronic odor stimulation was limited to either no change or small increases in PN sensitivity, with the effects of olfactory plasticity most significant for the encoding of weak odor stimuli.

## PN coding of odor mixtures is unaffected by chronic odor exposure

So far, we have evaluated PN odor responses using monomolecular odor stimuli, chosen because some activate only a single ORN class when presented at lower concentrations. We began with this approach so that the presynaptic source of odor-evoked input with respect to each PN type was unambiguous; however, typical odors activate multiple ORN classes (*de Bruyne et al., 2001*; *Hallem and Carlson, 2006*). Thus, we next investigated how chronic odor exposure impacts the coding of typical odor stimuli that elicit mixed direct and indirect synaptic input to PNs.

As before, we recorded from VM7 PNs in flies chronically exposed to E2-hexenal ($10^{-7}$), 2-butanone ($10^{-4}$), or solvent (*Figure 5A and D*). We mixed a fixed concentration of pentyl acetate ($10^{-3}$), a broadly activating odor that drives activity in many ORN types (but does not activate pb1a, the VM7 ORN), with increasing concentrations of 2-butanone, the odor that elicits direct activity in VM7 (*Olsen et al., 2010*; *Nagel and Wilson, 2011*). Overall, blending pentyl acetate with 2-butanone reduced VM7 PN responses, as compared to their response to 2-butanone alone (*Figures 5B and 2H* and *Figures 5E and 3B*). Such mixture inhibition is well understood to be a consequence of lateral GABAergic inhibition elicited by activity in non-VM7 ORNs (*Olsen et al., 2010*; *Olsen and Wilson, 2008*). Whereas responses of VM7 PNs to direct excitation (driven by 2-butanone) were modestly enhanced in 2-butanone exposed flies (*Figure 2H–I*), VM7 responses to mixed direct and indirect input (driven by blends of 2-butanone and pentyl acetate) were similar in control and 2-butanone exposed flies (*Figure 5B–C*). This effect was observed across a wide range of concentrations of 2-butanone, each blended with a fixed concentration of pentyl acetate. Similar results were observed when we recorded

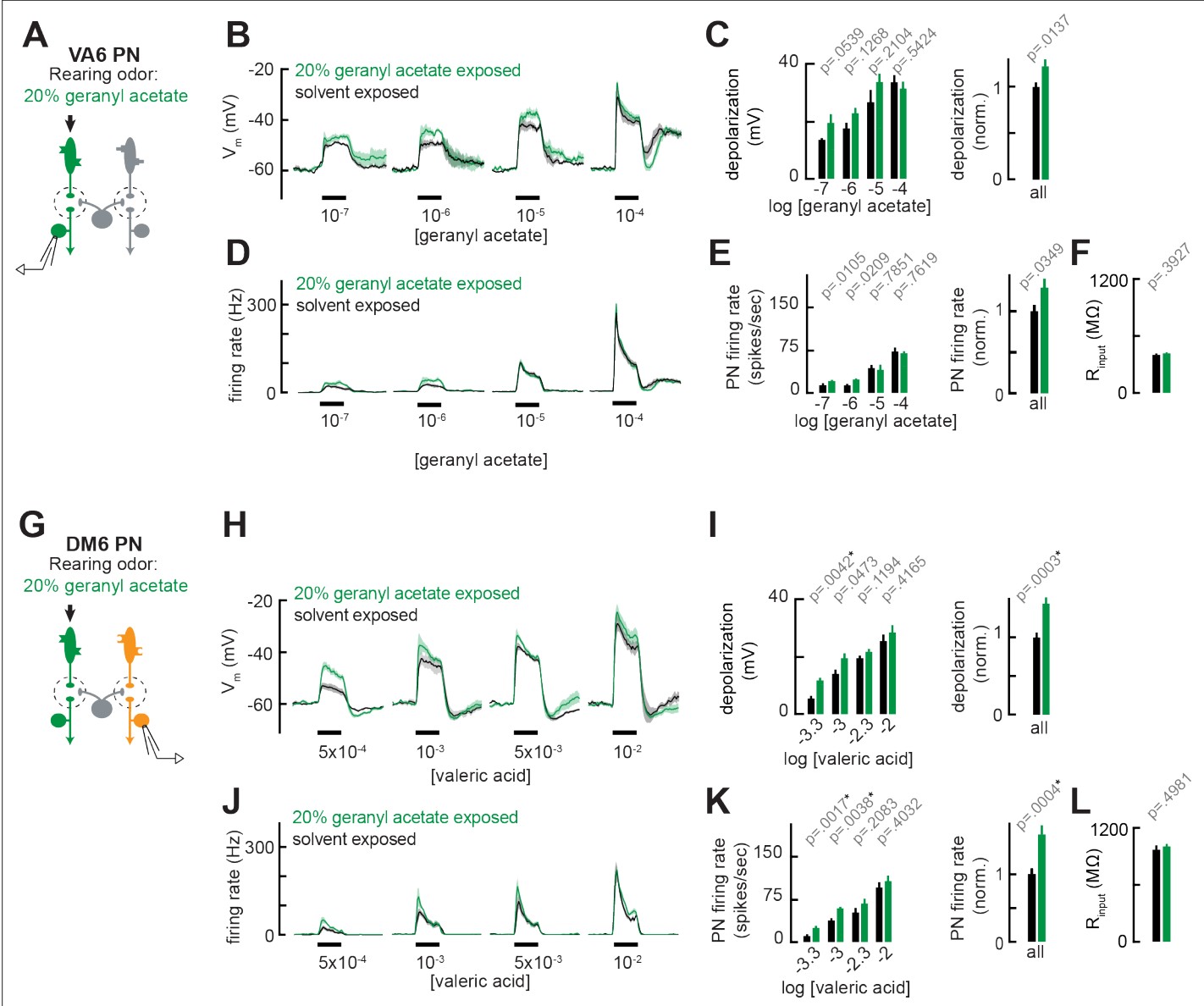

**Figure 4.** Chronic exposure to intense monomolecular odor can mildly enhance projection neuron (PN) responses to weak odor stimuli in PNs that receive either direct or indirect input from chronically active olfactory receptor neurons (ORNs). (**A**) Schematic of experimental setup for (**B–F**). Recordings were established from VA6 PNs receiving direct presynaptic input from the ORN class (ab5a) chronically activated by the rearing odor (geranyl acetate, 20%), n=5–6. (**B**) Odor-evoked depolarization in VA6 PNs in response to varying concentrations of geranyl acetate from flies chronically exposed to 20% geranyl acetate or solvent (paraffin oil). (**C**) *Left:* mean odor-evoked depolarization to each stimulus in (**B**) in the 500 ms after nominal stimulus onset. *Right:* mean normalized odor-evoked depolarization across all stimuli. Within each stimulus, responses were normalized to the mean odor-evoked depolarization in the solvent-exposed group. (**D**) Peristimulus time histograms of odor-evoked spiking responses in VA6 PNs from (**B**). (**E**) *Left:* mean odor-evoked firing rates to each stimulus from (**D**) in the 500 ms after nominal stimulus onset. *Right:* mean normalized odor-evoked firing rates across all stimuli, computed analogously as in (**C**). (**F**) Mean input resistance of VA6 PNs recorded from 20% geranyl acetate- or solvent-exposed flies. (**G**) Experimental setup for (**H–L**). Recordings were established from DM6 PNs that receive indirect activity from the ORN class (ab5a) chronically activated by the rearing odor (geranyl acetate, 20%), n=5–6 cells. (**H–L**) As in (**B–F**), but for DM6 PNs in response to varying concentrations of valeric acid from flies chronically exposed to 20% geranyl acetate or solvent (paraffin oil).

The online version of this article includes the following source data for figure 4:

**Source data 1.** Source data for *Figure 4B–C, D–E, H–I and J–K*.

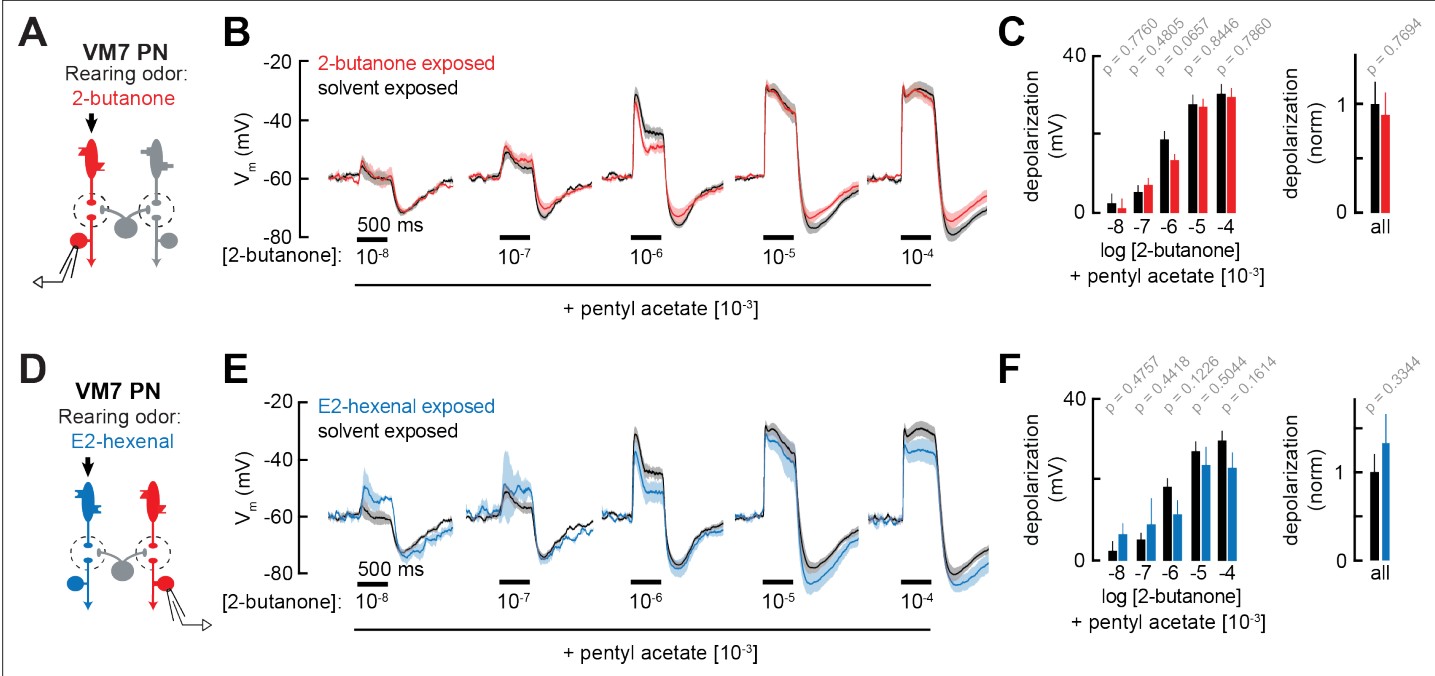

**Figure 5.** Projection neuron (PN) responses to odor mixtures are unaffected by chronic activation of direct or indirect olfactory receptor neuron (ORN) inputs. (**A**) Experimental setup for (**B–C**), which is the same as in *Figure 2G–J*. Recordings were established from VM7 PNs that receive direct input from the ORNs (pb1a) chronically activated by the rearing odor (2-butanone, $10^{-4}$), n=3–7 cells. (**B**) Odor-evoked depolarization in VM7 PNs from 2-butanone- or solvent-exposed flies to binary mixtures composed of increasing levels of 2-butanone ($10^{-8}$ through $10^{-4}$) blended with a fixed concentration of pentyl acetate ($10^{-3}$). (**C**) *Left:* mean odor-evoked depolarization to each stimulus in (**B**) in the 500 ms after nominal stimulus onset. *Right:* mean normalized odor-evoked depolarization across all stimuli. Within each stimulus, responses were normalized to the mean odor-evoked depolarization in the solvent-exposed group. (**D**) Experimental setup for (**E–F**), which is the same as in *Figure 3A–E*. Recordings were established from VM7 PNs which receive indirect activity from the ORNs (ab4a) chronically activated by the rearing odor (E2-hexenal, $10^{-7}$), n=3–7 cells. (**E–F**) Same as in (**B–C**) but for VM7 PNs from E2-hexenal or solvent-exposed flies. All plots are mean ± SEM across flies (one cell/fly) in each experimental condition. p-Values are as described in *Figure 2*. See *Supplementary file 1* for the full genotype and number of flies in every condition.

The online version of this article includes the following source data and figure supplement(s) for figure 5:

**Source data 1.** Source data for *Figure 5B–C and E–F*.

**Figure supplement 1.** Local neuron (LN) innervation of antennal lobe glomeruli is unchanged by chronic excitation of a single olfactory receptor neuron (ORN) class.

**Figure supplement 1—source data 1.** Source data for *Figure 5—figure supplement 1B-I*.

odor mixture responses from VM7 PNs in E2-hexenal exposed flies (*Figure 5D*), which received chronically elevated indirect activity. Whereas chronic exposure to E2-hexenal altered VM7 PN responses to 2-butanone (*Figure 3B–C and E*), VM7 PN responses to odor mixtures of 2-butanone and pentyl acetate were indistinguishable between E2-hexenal and solvent-exposed flies (*Figure 5E–F*). These observations suggest that lateral inhibition may also be impacted by chronic odor exposure, such that odor-evoked input elicits more inhibition to counter modest increases in PN excitation. In this way, stable PN responses are maintained to the most typical odors, which activate many ORs.

When we examined the anatomy of the LN network, however, we observed that it was not grossly affected by chronic odor exposure. Levels of innervation of individual olfactory glomeruli by the neurites of large subpopulations of inhibitory LNs (iLNs; measured as the ratio of iLN neurites to total synaptic neuropil) were largely unchanged by chronic odor exposure (*Figure 5—figure supplement 1A–C*). Unexpectedly, in flies chronically exposed to E2-hexenal only, many glomeruli tended to be smaller in volume than their counterparts in solvent-exposed flies (*Figure 5—figure supplement 1E*, G). Similar trends, however, were not observed in parallel experiments where flies were chronically exposed to 2-butanone (*Figure 5—figure supplement 1D*). Thus, chronic exposure to some, but not all, odors can elicit mild anatomical perturbations in the olfactory circuit. In contrast with previous reports (*Das et al., 2011*; *Sachse et al., 2007*; *Devaud et al., 2001*), however, under our

odor exposure conditions, changes in glomerular volume were not glomerulus-specific but extended globally beyond the chronically active glomerulus.

## Chronic activation of ORNs does not alter their odor response properties

Prior studies have suggested that chronic odor exposure increases the sensitivity of ORNs (*Chakraborty et al., 2009*; *Iyengar et al., 2010*). We wondered if, in flies chronically exposed to some odors, heightened PN responses to weakly activating stimuli (which elicit little lateral inhibition) might stem directly from changes in ORN activity. Heightened ORN sensitivity in odor-exposed flies might not be apparent in PN responses to stronger odors if circuit mechanisms such as lateral inhibition, which grow with stimulus strength, were acting to compensate changes in feedforward excitation.

To evaluate how chronic odor exposure affects ORN sensitivity, we exposed flies to E2-hexenal ($10^{-7}$) or 2-butanone ($10^{-4}$) as before and recorded extracellular activity from the ORN classes selectively activated by each odor stimulus (*Figure 6A and D*; see Materials and methods). We observed that chronic activation of either ORN type – ab4a ORNs in E2-hexenal exposed flies or pb1a ORNs in 2-butanone exposed flies – did not significantly impact spontaneous (*Figure 6—figure supplement 1E–G*) or odor-evoked firing rates across a wide range of stimulus concentrations (*Figure 6B–C and E–F*, *Figure 6—figure supplement 1A–B*), including those lower concentrations which elicited enhanced responses in postsynaptic DL5 or VM7 PNs (*Figure 2B–E and H–I*). In addition, we evaluated pb1a ORN (presynaptic to VM7) responses in E2-hexenal exposed flies (*Figure 6G*) because VM7 PN responses to odor were enhanced in this condition compared to controls (*Figure 3B–C*). These experiments showed that pb1a odor responses were largely unaffected by E2-hexenal exposure (*Figure 6H–I*). Although we observed a small decrease in response to 2-butanone at a concentration of $10^{-4}$, this difference did not consistently trend at nearby concentrations ($10^{-5}$ or $10^{-3}$) and did not reach statistical significance after correction for multiple comparisons (*Figure 6—figure supplement 1C*).

We next considered the possibility that small changes in ORN firing rate might not be resolvable in extracellular recordings from individual neurons but that the high convergence of ORNs onto PNs could amplify small differences in ORN firing into a measurable increase in PN response. Therefore, we used functional imaging to measure the population response of all ab4a ORNs in the DL5 glomerulus (*Figure 6J–K*), where the axon terminals of dozens of ab4a ORNs (*Grabe et al., 2016*) converge in a small physical volume (~200 µm³). We expressed the genetically encoded calcium indicator GCaMP6f in ab4a ORNs under the control of the Or7a-Gal4 promoter. We then chronically exposed these flies to either E2-hexenal or solvent and used two-photon microscopy to record odor-evoked ORN calcium signals in the DL5 glomerulus (*Figure 6K*). We found that population imaging of ORN terminals had comparatively higher sensitivity for detecting odor responses, demonstrated by the ability to resolve odor-evoked activity in ab4a ORNs in response to E2-hexenal at a concentration of $10^{-10}$ (*Figure 6L–M*), responses which are not detectable by extracellular recordings from individual ORNs (*Figure 6B–C*). However, functional imaging showed that odor-evoked responses in ab4a ORN terminals in glomerulus DL5 were indistinguishable between E2-hexenal exposed and control flies across the entire range of odor concentrations tested (*Figure 6L–M*, *Figure 6—figure supplement 1D*). Taken together, these results indicate that ORN odor responses are unaffected by perturbations in the odor environment that drive over a million additional spikes in each ORN over the course of 2 days of exposure. They also imply that the PN plasticity we observe likely stems from central cellular or circuit mechanisms, rather than from changes at the periphery.

## The role of central circuit mechanisms in olfactory plasticity

We next investigated several central mechanisms that might contribute to olfactory plasticity. First, we asked whether chronic odor exposure changes the intrinsic cellular excitability of PNs. Comparisons of the input resistance of PNs recorded in odor-exposed and control flies showed that postsynaptic input resistance was unaltered by chronic exposure to any of the odors in our study, regardless of odor concentration or whether PNs received direct or indirect chronic activation (*Figures 2F, J, P, 3D, L, 4F and L*). Consistent with these observations, *f-I* curves directly measuring the firing rate of deafferented DL5 PNs in response to current injection at the soma (*Figure 7A*) were indistinguishable between control and E2-hexenal exposed flies (*Figure 7B–C*; see also *Figure 7—figure supplement 1A–B*).

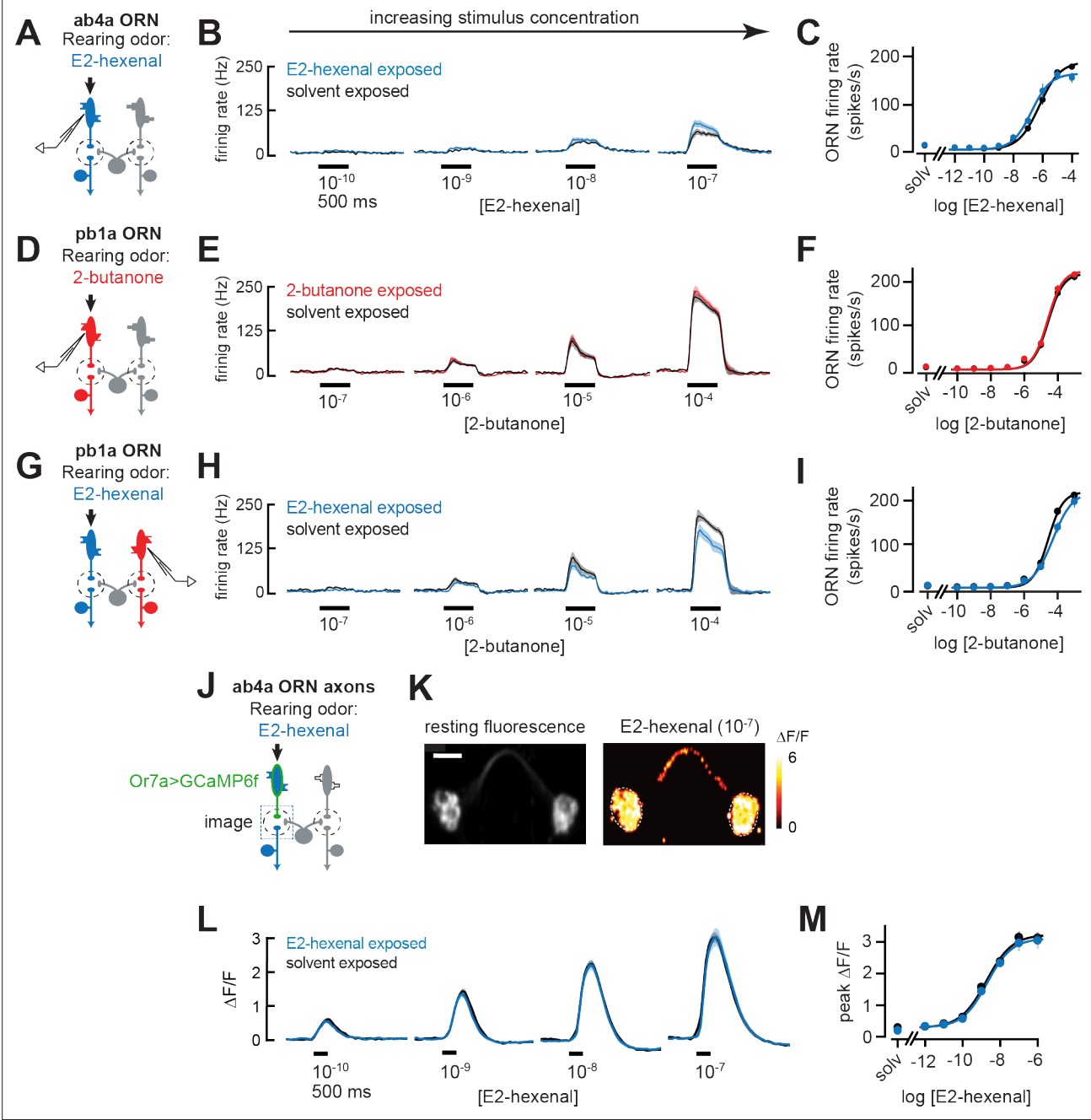

**Figure 6.** Odor responses in olfactory receptor neurons (ORNs) are unaffected by chronic odor exposure. (**A**) Experimental setup for (**B–C**). Single-sensillum recordings (SSR) were established from ab4a ORNs, which are directly excited by the rearing odor E2-hexenal ($10^{-7}$), n=6–12 cells. (**B**) Peristimulus time histograms of odor-evoked spiking in ab4a ORNs in response to varying concentrations of E2-hexenal from flies chronically exposed to E2-hexenal or solvent (paraffin oil). (**C**) Response curve of mean baseline-subtracted ab4a firing rates (calculated over the 500 ms window of stimulus presentation) to varying concentrations of E2-hexenal in E2-hexenal- or solvent-exposed flies. The concentration-response curve includes responses from (**B**), as well as measurements at additional stimulus concentrations. solv, solvent (paraffin oil). (**D**) Experimental setup for (**E–F**). SSR recordings from pb1a ORNs, which are directly excited by the rearing odor 2-butanone ($10^{-7}$), n=6–11 cells. (**E–F**) Same as (**B–C**) but for pb1a ORNs responding to varying concentrations of 2-butanone in flies chronically exposed to 2-butanone or solvent. (**G**) Experimental setup for (**H–I**). SSR recordings from pb1a ORNs in flies chronically exposed to E2-hexenal ($10^{-7}$), a stimulus which directly excites ab4a ORNs, n=5–11 cells. (**H–I**) Same as (**B–C**), but for pb1a ORNs responding to varying concentrations of 2-butanone in flies chronically exposed to E2-hexenal or solvent. (**J**) Experimental setup for (**K–M**). GCaMP6f was expressed in ab4a ORNs under the control of *Or7a-Gal4*. Flies were chronically exposed to E2-hexenal ($10^{-7}$) or solvent, and odor-evoked calcium responses in ab4a ORN terminals were imaged in the DL5 glomerulus (dashed box) using two-photon microscopy, n=6–8 cells. (**K**) *Left*: maximum intensity projection of the imaging plane across the time series of an example stimulus presentation. *Right*: peak ΔF/F heat map from

*Figure 6 continued on next page*

*Figure 6 continued*

a single experiment evoked by a 500 ms pulse of E2-hexenal ($10^{-7}$) in a solvent-exposed fly, averaged across three stimulus presentations. Scale bar is 5 μm. (**L**) Time courses of change in fluorescence in ab4a ORN terminals elicited by varying concentrations of E2-hexenal in E2-hexenal- and solvent-exposed flies. (**M**) Response curve of mean peak ΔF/F responses to varying concentrations of E2-hexenal in E2-hexenal- or solvent-exposed flies. The concentration-response curve includes responses from (**L**), as well as measurements at additional stimulus concentrations. solv, solvent. All plots are mean ± SEM across flies in each experimental condition (one cell or antennal lobe/fly). Statistical analysis was as in *Figure 2* (see *Figure 6—figure supplement 1* for p-values); none of the comparisons in *Figure 6* between odor- and solvent-exposed groups are statistically significant at the α=0.05 level, with Bonferroni adjustment for multiple comparisons. See *Supplementary file 1* for the full genotype and number of flies in every condition.

The online version of this article includes the following source data and figure supplement(s) for figure 6:

**Source data 1.** Source data for *Figure 6B–C, E–F, H–I and L–M*.

**Figure supplement 1.** Statistical analysis of spontaneous and odor-evoked olfactory receptor neuron (ORN) firing rates in odor- and solvent-exposed flies.

These results indicate that the intrinsic excitability of PNs is unaltered by chronic odor exposure and does not account for the increase in PN sensitivity to weak odors.

Next, we asked whether ORN-PN synaptic strength is impacted by chronic odor exposure. In each glomerulus, many axon terminals from the same ORN class synapse onto each uniglomerular PN and each ORN communicate with each PN via multiple active zones (*Horne et al., 2018*; *Kazama and Wilson, 2008*; *Rybak et al., 2016*; *Tobin et al., 2017*). We refer to the combined action of all the neurotransmitter release sites between a single ORN and a PN as a unitary ORN-PN synapse. To measure the strength of a unitary synaptic connection between ab4a ORNs and DL5 PNs, we adapted a previously established minimal stimulation protocol (*Kazama and Wilson, 2008*) for use with optogenetic-based recruitment ORN activity. We expressed the channelrhodopsin variant Chrimson (*Klapoetke et al., 2014*) in all ab4a ORNs, driven from the Or7a promoter (*Couto et al., 2005*), acutely severed the antennal nerve, and stimulated ORN terminals with wide-field light delivered through the imaging objective. Concurrently, we monitored synaptic responses in DL5 PNs using targeted whole-cell recordings in voltage clamp mode (*Figure 7D*).

We employed a minimal stimulation protocol to isolate unitary excitatory postsynaptic currents (uEPSCs) evoked by single presynaptic ORN spikes. Stimulation with very low levels of light elicited no synaptic response in the PN (*Figure 7E*). As the power density was gradually increased, trials of mostly failures were interspersed with the abrupt appearance of an EPSC in an all-or-none manner. Further ramping the light in small increments had no effect on the amplitude of the EPSC in the PN, until a power density was reached where the EPSC amplitude abruptly doubled, as compared to the amplitude of the initially recruited EPSC (*Figure 7E*). Light-evoked EPSCs were dependent on providing flies with the rhodopsin chromophore all-trans-retinal (ATR) in their food; PNs from flies raised on non-ATR supplemented food displayed no light-evoked responses (data not shown). The step-like profile of EPSC amplitudes as a function of power density likely reflects the discrete recruitment of individual ORN axon fibers with increasing stimulation. In particular, the sharp transition from mostly failures to a reliably evoked current is consistent with the response arising from the activation of a single ORN input. The time from the onset of light stimulation to the evoked uEPSC was variable and averaged approximately ~23 ms±2.2 ms (SD; *Figure 7—figure supplement 1C*), similar to the distribution of latencies to the first light-evoked ORN spike at comparable intensities (data not shown, and *Jeanne and Wilson, 2015*).

Using this optogenetic-based ORN recruitment method, the amplitude (~40 pA), rise time (~2 ms), and half decay time ($t_{1/2}$ ~7 ms) of uEPSCs in DL5 PNs (*Figure 7F*) in the control condition were similar to previous measurements made using conventional electrical stimulation of the antennal nerve (*Kazama and Wilson, 2008*), confirming this method for measuring unitary ORN-PN synapse properties.

We used this method to record uEPSCs from DL5 PNs in flies chronically exposed to either solvent or E2-hexenal ($10^{-7}$), the odor which directly activates the presynaptic ORNs. To compare recordings across different trials, individual uEPSCs from each condition were aligned by their peaks and averaged. Average DL5 uEPSC amplitudes and response kinetics were indistinguishable between solvent and E2-hexenal exposed flies (*Figure 7F–G*). This result shows that ORN-PN strength is unchanged by chronic odor exposure and is unlikely to account for enhanced DL5 PN responses to weak stimuli in E2-hexenal exposed flies.

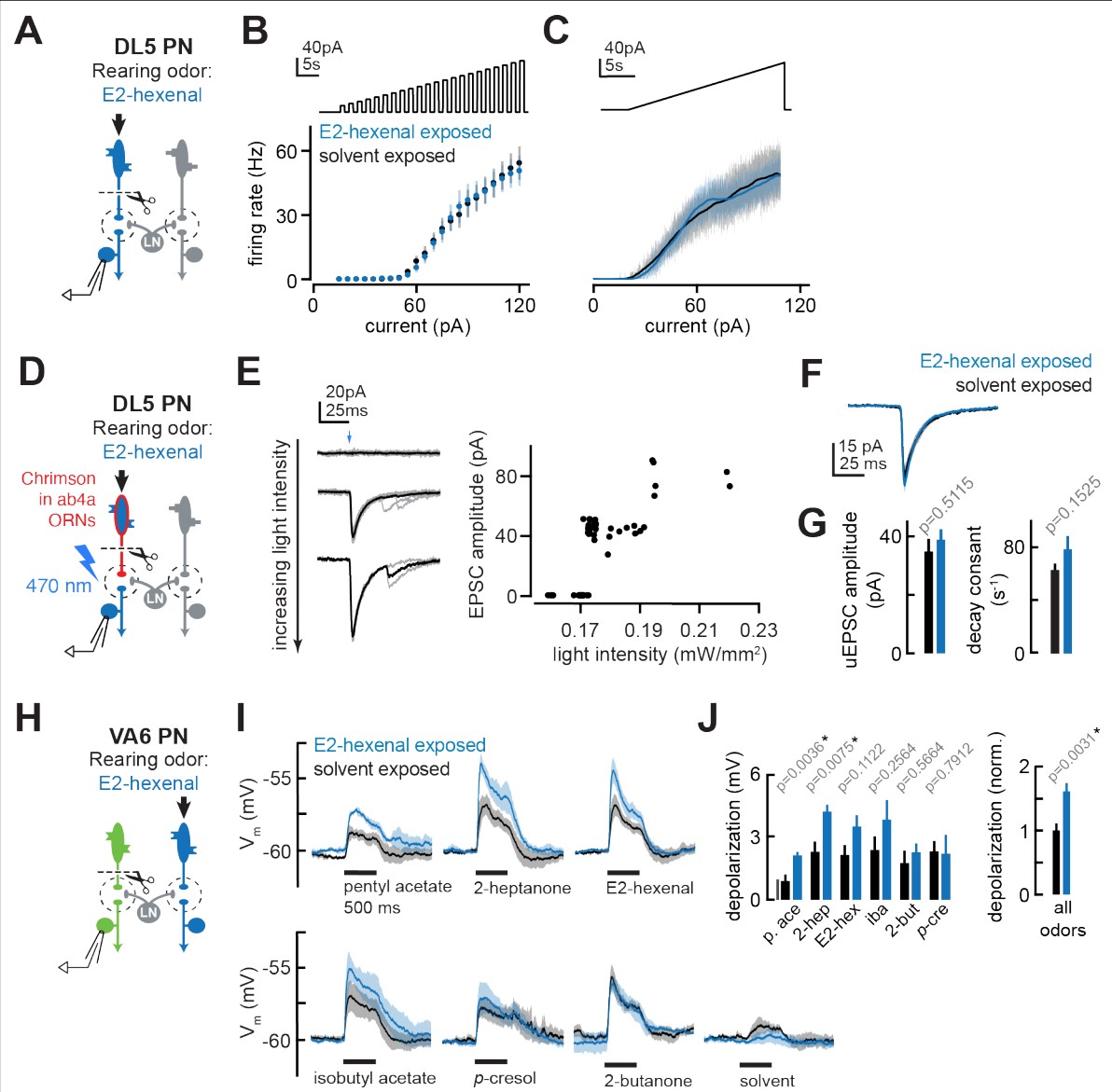

**Figure 7.** The effect of chronic olfactory receptor neuron (ORN) activation on projection neuron (PN) intrinsic properties, ORN-PN synapse strength, and lateral excitation in the antennal lobe. (**A**) Experimental setup for (**B–C**). Recordings were established from DL5 PNs, which receive direct presynaptic input from the ORN class (ab4a) chronically activated by the rearing odor (E2-hexenal, $10^{-7}$), n=4 cells. Immediately prior to recording, PNs were deafferented by bilateral transection of the antennal nerves. (**B**) f-I curve of DL5 PNs from E2-hexenal- and solvent-reared flies plotting firing rates elicited by increasing levels of current injection (1 s pulses delivered at an interpulse interval of 1 s and increasing with a step size of 5 pA). (**C**) Same as (**B**) but plotting firing rates elicited by injection of a slow triangular current ramp (4.5 pA/s). Firing rate was calculated in 50 ms bins with 25 ms overlap. (**D**) Experimental setup for measurement of synaptic strength between ab4a ORNs and DL5 PNs in (**E–G**). Flies expressing the channelrhodopsin CsCrimson in ab4a ORNs under the control of *or7a-Gal4* were chronically exposed to E2-hexenal ($10^{-7}$) or solvent. Immediately prior to the experiment, PNs were deafferented by bilateral transection of the antennal nerves. Recordings were established from DL5 PNs, and unitary excitatory postsynaptic currents (EPSCs) were elicited in PNs by light-based minimal stimulation of presynaptic ORN terminals. n=5–7 cells. (**E**) A minimal stimulation protocol recruits unitary EPSCs. *Left*: EPSCs recorded in a DL5 PN (from a solvent-exposed fly) in response to increasing levels of light-based (488 nm) ORN stimulation (blue arrow). Individual trials are in gray; the average of all trials at a given light intensity is in black. *Right*: EPSC amplitude as a function of light intensity. Each dot represents the peak EPSC amplitude from a single trial. As light intensity is gradually increased, an evoked EPSC appears abruptly, and its amplitude remains constant as the light intensity is further increased. This range (~0.17–0.19 mW/mm²) likely corresponds to recruitment of an action potential in a single ab4a ORN axon presynaptic to the DL5 PN. As the level of light driven ORN stimulation further increases, the amplitude of the evoked EPSC suddenly doubles, likely reflecting the recruitment of a second axon. (**F**) Mean unitary EPSC recorded in DL5 PNs from E2-hexenal- or solvent-exposed flies. (**G**) Mean unitary EPSC amplitude (left) and decay rate (right) in DL5 PNs from E2-hexenal- or solvent-exposed flies. (**H**) Experimental setup for (**I–J**). Recordings were established from VA6 PNs in flies chronically exposed to E2-hexenal or solvent. Immediately prior

*Figure 7 continued on next page*

*Figure 7 continued*

to recording, PNs were deafferented by bilateral transection of the antennal nerves. Odors stimulate intact ORNs located in the palps and recruit lateral input to VA6 PNs (which normally receive direct input from ORNs in the antenna). n=3–5 cells. (**I**) Odor-evoked depolarization in deafferented VA6 PNs elicited by the indicated stimuli in flies chronically exposed to E2-hexenal or solvent (paraffin oil). Odors were presented at $10^{-2}$ dilution. (**J**) *Left:* mean odor-evoked depolarization to each stimulus in (**I**) in the 500 ms after nominal stimulus onset. *Right:* mean normalized odor-evoked depolarization across all stimuli. Within each stimulus, responses were normalized to the mean odor-evoked depolarization in the solvent-exposed group. All plots are mean ± SEM across flies in each experimental condition (one cell/fly). p-Values are as described in *Figure 2*. See *Supplementary file 1* for the full genotype and number of flies in every condition.

The online version of this article includes the following source data and figure supplement(s) for figure 7:

**Source data 1.** Source data for *Figure 7B–C, F–G and I–J*.

**Figure supplement 1.** Projection neuron (PN) intrinsic properties and latency of light-evoked excitatory postsynaptic currents (EPSCs).

Finally, we considered the possibility that local lateral excitatory connections play a role in PN plasticity triggered by chronic odor exposure. Olfactory glomeruli in the antennal lobe are densely interconnected by a network of lateral excitatory LNs (eLNs) that signal globally via electrical synapses to boost PN excitability (*Olsen et al., 2007*; *Root et al., 2007*; *Shang et al., 2007*; *Yaksi and Wilson, 2010*). The effects of lateral excitation on PN responses are most significant in the regime of weak odor stimuli (*Shang et al., 2007*; *Yaksi and Wilson, 2010*). Lateral excitatory input to PNs is most directly measured by removing the source of direct input to the PN and recording PN activity while stimulating ORNs that directly excite other glomeruli (*Olsen et al., 2007*; *Root et al., 2007*; *Shang et al., 2007*). The fly has two sets of olfactory organs – the antennae and the palps – and each ORN class resides in one or the other but never both. We chronically exposed flies to either solvent or E2-hexenal ($10^{-7}$), bilaterally removed the antennae, and established recordings from VA6 PNs while stimulating the palps with odor (*Figure 7H*). Since VA6 PNs receive direct input from ORNs located in the antenna, VA6 PN odor responses recorded in this configuration stem from lateral (indirect) input that originates from ORNs located in the palp.

The stimulus panel for this experiment was composed of odors that broadly activate many ORN classes, including those housed in the palps. As previously shown (*Olsen et al., 2007*; *Yaksi and Wilson, 2010*), different odors elicit differing but characteristic amounts of lateral excitation in VA6 PNs (*Figure 7I*). Many, but not all, odors evoked increased lateral excitation in VA6 PNs from E2-hexenal exposed flies, as compared to solvent-exposed flies (*Figure 7I–J*). To pool our measurements of lateral excitatory responses in VA6 PNs across stimuli, we normalized the amount of PN membrane depolarization elicited by each odor in E2-hexenal exposed flies to the average amount it elicited in solvent-exposed flies. This analysis confirmed that the average amount of odor-evoked lateral excitation across all stimuli was increased in E2-hexenal exposed flies, as compared to solvent-exposed controls (*Figure 7J*). These results suggest that chronic odor exposure increases the overall strength of global excitatory coupling among glomeruli in the antennal lobe after chronic odor exposure, which may contribute to the heightened sensitivity of PNs to weak odors.

## Discussion

We found that strong, persistent activation of a single class of ORNs triggers only limited plasticity at early stages of olfactory processing in *Drosophila*. Chronic exposure to monomolecular odors at concentrations that can be found in the natural world elicited modest increases in the olfactory sensitivity of some PNs. Qualitatively similar effects on PN odor coding were observed even when exposing flies to high concentrations of a monomolecular odor, geranyl acetate. When present, experience-dependent plasticity in PNs mostly affected the encoding of weak odor stimuli, whereas responses to stronger odors, which more strongly recruit local inhibition, were largely unaffected. Many elements of the antennal lobe circuit, including ORN sensitivity, PN intrinsic properties, and ORN-PN synapse strength, were unaffected by chronic ORN activation. Plasticity triggered by chronic ORN activity was observed not only in PNs corresponding to the glomerulus that receives direct input from the chronically active ORNs but also in PNs corresponding to other glomeruli, indicating that experience-dependent plasticity in PNs is not glomerulus- or odor-specific. This result implicates lateral interactions between glomeruli as having a role in olfactory plasticity, consistent with our observation that odor exposure can boost the level of lateral excitatory coupling between some PNs. Thus, even in

odor environments that elicit unusually high, long-lasting levels of activity in a single ORN class, the representation of odors in the antennal lobe is mostly stable. Chronic odor stimulation results in either no or mild changes in PN responses, which sensitize PNs to weak stimuli and extend the lower range of stimulus intensities dynamically encoded by the PN. Reduced PN responses were not observed in response to chronic excitation in any condition we tested, inconsistent with the hypothesis that PN olfactory codes adjust to the frequency with which specific odors are encountered in the environment.

## Chronic exposure to odors elicits limited plasticity in PN odor responses

Chronic activation of at least two ORN classes by odors at naturalistic concentrations elicited modest increases in the odor sensitivity of the cognate PNs (DL5 and VM7) receiving direct presynaptic input from each. A third PN type (VA6) was unaffected by exposure to lower naturalistic concentrations of odor (*Figure 2*). These results contrast with prior studies in flies in which chronic exposure to mono-molecular odors delivered at high concentrations selectively reduced the sensitivity of PNs activated by these odors (*Das et al., 2011*; *Sachse et al., 2007*). We considered the possibility that exposure to higher concentrations of the odor was necessary to elicit PN plasticity; however, we found that chronic sustained exposure to high concentrations of geranyl acetate (from a 20% v/v source) had a similar effect on VA6 PN odor responses as exposure to geranyl acetate at more than a 1000-fold lower concentration. This result is expected because VA6 PNs are near maximally activated (~300 Hz peak firing rate) at the naturalistic concentration ($10^{-4}$) used, and further increasing the concentration of odor does not drive substantially more activity in VA6 PNs. These findings show that the commonly accepted idea that chronic odor exposure reduces the sensitivity of PNs processing the familiar odor is, at the very least, not true for all odors and glomeruli.

Establishing the direction of olfactory plasticity evoked in response to elevated levels of olfactory input is important, as it would point toward differing functional consequences of plasticity for olfactory coding. A prior study observed that sustained, chronic exposure of flies to 1% geranyl acetate weakly enhanced VA6 PN responses (*Kidd et al., 2015*), a result which the current study confirms and extends to additional odors and glomeruli. Another analogous study in mice found that odor-evoked mitral cell activity in the olfactory bulb was modestly increased after chronic odor exposure to intense monomolecular odors (*Liu and Urban, 2017*). In both prior studies, odor-evoked activity was measured using functional calcium imaging, suggesting that the use of electrophysiology to measure olfactory activity in our study does not account for the difference in outcomes. Since methodological details differ between any two studies, we do not know whether chronic odor stimulation can reduce PN sensitivity in some specific contexts. For instance, we cannot rule out the possibility that chronic excitation of different glomeruli has different outcomes, such that PN sensitivity is adjusted according to specific rules useful for each individual odor and glomerulus. Much of the prior work has focused on the V glomerulus, which processes the important environmental cue carbon dioxide and could be subject to a different form of plasticity. Because of the position of PN cell bodies in the antennal lobe, evaluating V PN odor responses using whole-cell recordings is technically challenging. However, in PNs corresponding to the three different glomeruli we did study, each activated by different odors, chronic excitation was consistently observed to boost responses to weak odors in most cases, and reduced PN responsiveness was never observed.

## The use of low, naturalistic odor concentrations for studying olfaction

In this study, flies were exposed to periodic 1 s pulses of odor at estimated concentrations of ~10 ppb to ~10 ppm in air (see Materials and methods). Though the overrepresented odor narrowly activated a single olfactory channel, it was delivered to flies living in an active culture containing cornmeal food, yeast, and other flies. As such, the olfactory circuit is expected to be broadly active during the period of chronic odor exposure. Our goal with this experimental design was to drive a robust difference in levels of neural activity in a single ORN type compared to control animals, while still maintaining animals in an olfactory environment that could be plausibly encountered in the natural world.

Nearly all prior studies investigating olfactory plasticity, in flies or rodents, chronically excite the olfactory system using odors delivered at comparatively higher concentrations, generally ranging from ~$10^3$ to ~$10^5$ ppm in air (*Das et al., 2011*; *Sachse et al., 2007*; *Liu and Urban, 2017*; *Devaud et al., 2001*; *Chodankar et al., 2020*; *Wang et al., 1993*). Such stimuli are unlikely to be found in

natural odor sources; for comparison, headspace concentrations of the most abundant small ester, alcohol, and aldehyde volatiles common in fruit odor sources typically range from ~1 ppb to ~10 ppm in air (e.g. see *Boschetti et al., 1999*; *Farneti et al., 2017*; *Jordán et al., 2001*). Indeed, for many volatile organic compounds, prolonged exposure at concentrations that exceed ~$10^3$ ppm is considered hazardous to human life or health (NIOSH *CDC, 2019*). Since rates of odor-evoked firing in many olfactory neurons are saturated at concentrations well below these intense concentrations, it may be preferable, when possible, to use odor stimuli within the concentration range likely to be encountered in the natural evolutionary history of the animal. However, we note that, though it was not the subject of this study, understanding how exposure to very intense odors impacts olfactory behavior and function is an important priority since both animals and humans frequently encounter such situations in modern industrialized environments (*Steinemann, 2016*; *Wolkoff and Nielsen, 2017*; *Wypych, 2017*).

## Stimulus-selective versus global plasticity in sensory circuits

Another important difference between our results and some prior work is that the mild olfactory plasticity we observed did not selectively occur only in the glomerulus receiving direct input from the chronically activated ORN class. Some prior studies showed that the effects of chronic odor exposure on olfactory neuron responses and anatomical volume affect only specific glomeruli, although the direction of these effects varied between studies (*Das et al., 2011*; *Sachse et al., 2007*; *Devaud et al., 2001*; *Devaud et al., 2003*; *Chakraborty et al., 2009*). Most studies exposed flies to odors that broadly activate many ORN classes, complicating the interpretation of the degree of selectivity of olfactory plasticity. In cases where flies were exposed to an intense monomolecular odor that excited a single ORN class, for instance, $CO_2$-sensitive ORNs that project to the V glomerulus, olfactory responses of PNs receiving direct and indirect chronic excitation were not directly compared (*Das et al., 2011*; *Sachse et al., 2007*). Determining the degree to which PN plasticity is stimulus- and glomerulus-specific is significant because it affects the extent to which PN plasticity can account for stimulus-specific changes in olfactory behavior after chronic exposure, as well as whether PN plasticity can reshape the olfactory code to reflect the distribution of specific odors in the environment.

This study rigorously tested this hypothesis by growing animals in odor environments that selectively increased olfactory activity in a single ORN class and then evaluating the olfactory sensitivity of PNs that receive either direct or indirect activity from the chronically excited ORNs. Under our experimental conditions, we found that olfactory plasticity is nonselective and occurs broadly in many glomeruli. Chronic ORN excitation could trigger mild increases in odor responses both in PNs receiving direct excitation and PNs receiving indirect excitation from the chronically active ORN class.

Plasticity elicited by chronic indirect input did not impact all PNs equivalently. For example, chronic elevation of indirect input to VM7 PNs (by exposure to E2-hexenal) mildly increased VM7 PN sensitivity and also significantly increased levels of post-stimulus hyperpolarization in the cell. However, this latter effect was not observed in VA6 PNs or in DM6 PNs in flies exposed to 20% geranyl acetate, which exhibited normal post-stimulus dynamics (*Figure 3B and J*). These results suggest that olfactory plasticity differentially impacts PNs in different glomeruli, possibly due to how the circuit mechanisms underlying plasticity interact with the varying intrinsic biophysical characteristics of each PN type.

Our observation that chronic olfactory stimulation broadly affects many PN types is similar to that of a recent functional imaging study in mouse which found that chronic odor exposure in early postnatal life induced widespread, global enhancement of mitral cell excitability across the olfactory bulb (*Liu and Urban, 2017*). Chronic, persistent olfactory activity may function to adjust the overall gain or sensitivity of the circuit, especially in the weak stimulus regime. Such a widespread increase in excitability might reflect a form of generalized sensory enrichment that has been previously described in mammalian olfactory (*Mandairon et al., 2006b*; *Rochefort et al., 2002*), visual (*Beaulieu and Cynader, 1990*), and auditory (*Engineer et al., 2004*) systems.

## Mechanisms of olfactory plasticity in the antennal lobe

PN odor responses depend on the nonlinear integration of a complex set of inputs, which include feedforward excitation from ORNs, lateral excitation from cholinergic LNs (eLNs), and lateral inhibition from GABAergic LNs (iLNs; *Wilson, 2013*). In the antennal lobe, the strength of each of these inputs is stereotypical in each glomerulus (*Hong and Wilson, 2015*; *Kazama and Wilson, 2008*;

*Olsen et al., 2007*; *Yaksi and Wilson, 2010*). PN plasticity could theoretically arise from changes in any of these inputs, as well as changes in the intrinsic biophysical properties of the PN that impact signal integration. The observation that chronic focal activation of ORN input to a single glomerulus can elicit changes in odor coding in PNs belonging to other glomeruli suggests that olfactory plasticity is not glomerulus-autonomous and, at a minimum, likely involves local lateral networks which mediate information flow across glomeruli. Indeed, past studies in insects have implicated local inhibitory networks in mediating short- and long-timescale plasticity in the antennal lobe. For instance, repeated encounters with a given odor over short timescales (seconds to minutes) restructures PN activity to increase the reliability of PN representations of the odor (*Stopfer and Laurent, 1999*; *Franco and Yaksi, 2021*; computational models indicate this plasticity can be explained by facilitation of inhibitory connections in the antennal lobe; *Bazhenov et al., 2005*). In plasticity elicited by odor experience on long timescales (days), past work suggests glomerulus-specific plasticity is mediated by the strengthening of inputs from specific genetically defined subsets of iLNs (*Das et al., 2011*; *Sachse et al., 2007*; *Golovin and Broadie, 2016*). This mechanism assumes that patterns of activity in the neurites of iLNs, each of which ramify broadly in the vast majority of antennal lobe glomeruli, are selectively modified through an as-yet-undescribed process to regulate release sites in just one or a few glomeruli. Our observation that olfactory plasticity is not necessarily glomerulus-specific is consistent with the broadly innervating anatomical characteristics of both excitatory and iLNs.

Prior work in fly found that chronic exposure to esters increased the sensitivity of ORNs to these odors (*Chakraborty et al., 2009*; *Iyengar et al., 2010*), and multiple studies in rodents have concluded that chronic odor exposure evokes plasticity in ORN responses (*Wang et al., 1993*; *Cadiou et al., 2014*; *Cavallin et al., 2010*; *Kass et al., 2013*; *Santoro and Dulac, 2012*; *Watt et al., 2004*). We considered the possibility that large changes in ORN sensitivity were not reflected in PN odor responses due to compensatory adjustments in other parts of the antennal lobe circuitry. Direct measurements of how chronic odor exposure impacted ORN odor coding, PN intrinsic properties, ORN-PN synaptic strength, and lateral excitation revealed that, overall, most circuit properties were remarkably stable to a major perturbation in the flies' olfactory environment. For instance, the olfactory responses of ORNs in multiple glomeruli, as measured by single sensillum extracellular recordings and by population calcium imaging, were unaffected by strong, chronic excitation and were stable across a wide range of odor concentrations. Likewise, PN intrinsic properties and ORN-PN synaptic strength were similarly invariant to chronic odor exposure. However, we found that the strength of lateral excitatory coupling among glomeruli was increased in flies chronically exposed to the odor E2-hexenal. This result suggests a possible mechanism for the overall increased excitability of PNs in E2-hexenal exposed flies, particularly in the weak stimulus regime in which lateral excitation has the most impact (*Yaksi and Wilson, 2010*). It would be interesting to evaluate whether the strength of coupling of a given glomerulus into the lateral excitatory network, which varies across PN types (*Yaksi and Wilson, 2010*), relates to the magnitude of plasticity evoked by chronic excitation of that glomerulus. A change in the strength of lateral excitatory coupling between eLNs and PNs has also been previously implicated in the slow recovery of odor responses in chronically deafferented PNs (*Kazama et al., 2011*). Thus, the lateral excitatory network might serve as a substrate for olfactory plasticity in multiple contexts.

## Implications for interpreting odor experience-driven changes in olfactory behavior

Many studies in insect and in mammals have demonstrated that prior experience with an odor impacts how an animal subsequently responds to it. Establishing the directionality of olfactory plasticity – whether olfactory PNs respond less, more, or no differently to overrepresented odors in the environment – is important for understanding olfactory behavioral plasticity. Reports of reduced PN responses to overrepresented odors in the environment (*Das et al., 2011*; *Sachse et al., 2007*) were central to the interpretation of reduced behavioral aversion of flies toward familiar odors as a form of behavioral habituation. However, since nearly all studies use high concentrations of monomolecular odor (from a ~1–20% v/v source), which are nearly always aversive to animals, these experiments do not disambiguate whether reduced behavioral aversion reflects behavioral habituation to an aversive odor or increased attraction (or tolerance) to an aversive odor. In one study where flies were chronically exposed to monomolecular odors at lower concentrations (~0.01–1%) that are attractive to flies,

odor experience increased behavioral attraction toward the familiar odor (*Chakraborty et al., 2009*). In agreement, other work from our laboratory shows that chronic exposure of flies in early life to odors from natural sources increases behavioral attraction to these odors (*Dylla et al., 2023*). These observations argue against habituation being the dominant effect of chronic odor exposure on behavior since, if that were the case, reduced attraction to familiar attractive odors is expected.

Prior studies have suggested that experience-dependent modification of olfactory behavior in fly stems from odor-specific changes in the structure and function of the antennal lobe, which act to reduce PN sensitivity to frequent or abundant odors in the environment (*Das et al., 2011*; *Sachse et al., 2007*; *Devaud et al., 2001*; *Sudhakaran et al., 2012*). The overall stability of odor coding in PNs, the major output from the antennal lobe, in the face of significant perturbations in the odor environment that chronically alters the distribution of sensory input to the fly olfactory system, argues against this idea. Indeed, given that most typical odors are broadly encoded across many glomeruli, glomerulus-selective plasticity in the antennal lobe would seem to be an inefficient substrate for odor-specific plasticity since most individual glomeruli participate in the representation of many different odors.

If chronic exposure to odors in early life is reinterpreted as an increase in attraction or acceptance of familiar odors, rather than habituation toward them, our observation of stable PN responses in odor-exposed flies suggests that the neural mechanism responsible for behavioral plasticity likely acts downstream of the antennal lobe. One higher order olfactory area which receives antennal lobe output, the mushroom body, has been extensively studied for its role in associative learning (*Heisenberg, 2003*). Indeed, chronic odor exposure experiments are nearly always, by necessity, carried out in the presence of food, which may signal to flies a positive value of the environment and become associated with the odor. More work is needed to evaluate how behavioral plasticity elicited by chronic odor exposure may depend on additional features of the environment, for instance, if exposure occurs in a passive versus rewarding (or aversive) context.

## Implications for general principles of sensory plasticity

The stability of odor responses in early olfactory processing areas, even when challenged with persistent perturbations in the sensory environment, may reflect a more general design principle of sensory circuits. Even in mammalian nervous systems, which exhibit an overall higher degree of neural plasticity than insect systems, the function of early stages of sensory processing closer to the periphery is less dependent on normal sensory experience than later stages of cortical processing. For example, although normal visual experience is required for normal topographic maps, orientation selectivity, and direction selectivity in higher visual areas (*Espinosa and Stryker, 2012*; *Cang and Feldheim, 2013*; *Huberman et al., 2008*), the structure and function of retinal circuitry are much less impacted by abnormal visual experience (*D'Orazi et al., 2014*; *Elstrott and Feller, 2009*). Taking the case of direction selectivity as an example, whereas raising animals in the dark prevents the emergence of direction-selective responses in the primary visual cortex (*Li et al., 2006*; *Li et al., 2008*), direction-selective ganglion cells in the retina have mature responses at birth, and they have normal directional tuning, speed tuning, and anatomy in dark-reared animals (*Elstrott and Feller, 2009*; *Chan and Chiao, 2008*; *Elstrott et al., 2008*). Thus, in both insects and vertebrates, experience-independent processes, specified by developmental genetic programs, appear to dominate in determining the structure and function of early stages of sensory processing, with the role of sensory experience becoming more prominent in higher-order stages of processing.

Why might plasticity be limited in the early stages of sensory processing? Neural plasticity, like any form of phenotypic plasticity, comes at a cost. For instance, plasticity at the sensory periphery could be subject to an information acquisition cost, stemming from poor reliability or under-sampling of the stimuli being used to evaluate the statistical structure of the environment. Another potential cost of plasticity could arise from temporal mismatching, for instance, if the stimulus structure of the environment were to shift more rapidly than the timescale over which neural plasticity could be implemented. Generating a relatively stable representation of the world at the early stages of processing, which is invariant to local shifts in the stimulus environment, maybe the best strategy. The initial sensory representation is relayed to multiple higher-order processing areas, each of which may use the sensory information for different behavioral tasks. Neural plasticity acting at later stages of processing may

allow different downstream circuits to independently reformat the sensory representation in a way that best subserves its specialized function.

## Materials and methods

### Flies

*D. melanogaster* were raised on a 12:12 light:dark cycle at 25°C and 70% relative humidity on cornmeal/molasses food containing: water (17.8 l), agar (136 g), cornmeal (1335.4 g), yeast (540 g), sucrose (320 g), molasses (1.64 l), $CaCl_2$ (12.5 g), sodium tartrate (150 g), tegosept (18.45 g), 95% ethanol (153.3 ml), and propionic acid (91.5 ml). All experiments were performed in 2-day old female flies. The specific genotype of the flies used in each experiment is given in *Supplementary file 1*. The transgenes used in this study were acquired from the Bloomington *Drosophila* Stock Center (BDSC) or the Kyoto *Drosophila* Stock Center (DGGR), unless otherwise indicated. They have been previously characterized as follows: *NP3481-Gal4* (Kyoto:113297) labels DL5, VM7, and DM6 PNs (*Olsen et al., 2007*; *Hayashi et al., 2002*); *MZ612-Gal4* (II) (gift of L. Luo) labels VA6 PNs (*Marin et al., 2005*); *Or7a-Gal4*(KI) (gift of C. Potter) expresses Gal4 from the *Or7a* locus under the control of its endogenous regulatory elements (*Lin et al., 2015*); *UAS-CD8:GFP* (X) (RRID:BDSC_5136) and *UAS-CD8:GFP* (II) (RRID:BDSC_5137) express CD8-tagged GFP, which is targeted to the membrane, under Gal4 control (*Lee and Luo, 1999*); *20xUAS-IVS-CD8:GFP* (attP2) (RRID:BDSC_32194) expressed CD8-tagged GFP under Gal4 control (*Pfeiffer et al., 2010*); *UAS-brp.S-mStrawberry* (II) (gift of S. Sigrist) expresses a red fluorescent protein-tagged short-form of bruchpilot (*Fouquet et al., 2009*); *20xUAS-IVS-syn21-opGCaMP6f-p10* (su(Hw)attP5) (gift of B. Pfeiffer and D. Anderson) expresses codon-optimized GCaMP6f under Gal4 control (*Chen et al., 2013*); and *13xlexAop2-IVS-CsChrimson.mVenus* (attP40) (RRID:BDSC_55138) expresses a Venus-tagged red-shifted channelrhodopsin CsChrimson under lexA control (*Klapoetke et al., 2014*).

*Or7a-lexA* (III) flies were generated as follows. The *Or7a* promoter was PCR amplified from a bacterial artificial chromosome (RPCI-98 library, clone 39F18, BACPAC Resources) containing the *or7a* locus of *D. melanogaster* using primers 5′-ACCGCATCCCGATCAAGACACAC-3′ and 5′-TGATGGAC TTTTGACGCCTGGGAATA-3′. The *Or7a* promoter was inserted 5′ to *nlslexA::p65* using isothermal assembly in vector *pBPnlslexA::p65Uw*, replacing the ccdB cassette. The plasmid *pBPnlslexA::p65Uw* was a gift from G. Rubin (Addgene plasmid #26230, RRID:Addgene_26230). The final sequence of the construct was confirmed by Sanger sequencing, and transgenic flies were generated by site-specific integration into the VK00027 landing site (BestGene, Inc, Chino Hills, CA, USA). To verify the selectivity of the driver, *Or7a-lexA* was crossed to *13xlexAop2-mCD8:GFP* (RRID:BDSC_32205), and brains of the resulting progeny flies (2 days old) were dissected and immunostained with antibodies against GFP and nc82. GFP expression was observed selectively in ab4a ORN axons projecting to the DL5 glomerulus; no other signal was observed in the central brain.

### Chronic odor exposure

With the exception of *Figure 4*, all data were from flies that were chronically exposed to solvent or specific monomolecular odors while reared in standard fly bottles containing cornmeal/molasses food and sealed with modified cotton plugs through which two thin-walled stainless-steel hollow rods (~5 cm length and ~3.2 mm inner diameter) were tightly inserted, serving as an inlet and an outlet for airflow. The bottom of the cotton plug was lined with mesh (McMaster-Carr #9318T45) to prevent flies from entering the rods. The inlet port was fit with a luer connector for easy connection to the carrier stream; the outlet port was vented with loose vacuum suction.

The odor environment inside the bottle was controlled by delivering to the inlet of the bottle a stream of charcoal-filtered, humidified air (275 ml/min), with a small fraction of the air stream (odor stream, 25 ml/min) diverted into the headspace of a control vial filled with solvent (paraffin oil, J.T. Baker, VWR #JTS894-7) before it was reunited with the carrier stream (250 ml/min). Airflow rates were controlled using variable area valved flow meters (Cole-Parmer). In response to an external 5 V command, a three-way solenoid valve redirected the 25 ml/min odor stream from the headspace of the control solvent vial through the headspace of the vial containing diluted odor for 1 s. The diluted odor was continuously stirred using a miniature magnetic stir bar and stir plate (homebrewing.org). Delivery of the 1 s pulse of odor into the carrier stream was repeated every 21 s. Tygon tubing (E-3603)

was used throughout the odor delivery system, with the exception of a portion of the carrier stream where odor entered and the path from the odor vial to the input to the carrier stream, where PTFE tubing was used.

The stability of the amplitude of the odor pulse over the course of 24 hr was measured using a photoionization detector (200B miniPID, Aurora Instruments), with the sensor probe mounted at the center of a fresh fly bottle. Based on the observed rundown in the amplitude of the odor pulse (*Figure 1B*), the odor vial in the odor delivery system was swapped out every 12 hr for a fresh dilution of odor during chronic exposure experiment with flies. Under these conditions, the amplitude of the odor pulse was not expected to decrease more than ~20% at any point during the exposure period.

Flies were seeded in a fresh fly bottle at low density (~7–8 females). The evening prior to expected eclosion, any adult flies were removed from the bottle, and controlled odor delivery was initiated into the bottle. The next morning (day 0), newly eclosed flies were transferred into a fresh bottle, and controlled odor delivery was continued for another ~48 hr. Experiments were typically conducted on day 2, with the exception of those in *Figure 4*.

In *Figure 4*, flies were chronically exposed to a sustained high concentration of monomolecular odor from a stationary source placed in the bottle (*Das et al., 2011*; *Devaud et al., 2001*). The evening prior to eclosion, growth bottles were cleared of any adult animals, and a 2 ml glass vial containing 1 ml of geranyl acetate (20% v/v) was placed into the bottle. The vial opening was covered with a layer of porous mesh, held securely in place with a silicone o-ring fitted around the mouth of the vial. The growth bottle was sealed with a standard cotton plug with no inlets/outlets. The next morning, newly eclosed flies were transferred into a fresh bottle containing a fresh odor source, and thereafter, the odor source was renewed daily until day 5 when experiments were conducted.

Odor concentrations in this study are referred to by the v/v dilution factor of the odor in paraffin oil in the odor vial. For chronic odor exposure in *Figures 1–3* and *Figures 5–7*, flies were exposed to odors at dilution factors of $10^{-7}$ for E2-hexenal, $10^{-4}$ for 2-butanone, and $10^{-4}$ for geranyl acetate. Headspace concentrations were further diluted 1:11 in air prior to delivery to the fly bottle. Flies were chronically exposed to pulses of odor (in gas phase) that are estimated from published vapor pressure data at 25°C (*Kim et al., 2021*) to be ~1 ppb for E2-hexenal ($10^{-7}$), ~13,000 ppb for 2-butanone ($10^{-4}$), and ~4 ppb for geranyl acetate ($10^{-4}$). For chronic odor exposure in *Figure 4*, flies were exposed to geranyl acetate at a dilution factor of 0.2 corresponding to ~7,900 ppb in air.

The mean spontaneous firing rates of the PNs (DL5, VM7, and VA6) investigated in this study range from ~3–7 Hz, and each 1 s odor pulse delivered during chronic odor exposure elicits ~150 spikes in PNs (*Figure 1*). Thus, chronic odor exposure approximately doubles overall PN firing rates (from ~180–420 spikes/min to 690–930 spikes/min) and elicits ~1.3 million extra spikes in a specific PN type over the course of 2 days.

## Electrophysiological recordings
### PN recordings
Electrophysiological measurements were performed on 2-day-old female flies essentially as previously described (*Wilson et al., 2004*), except in *Figure 4*, where recordings were established from 5-day-old female flies. The rate of establishing successful recordings was significantly lower in *Figure 4*, approximately one in five attempts compared to the usual rate of approximately one in two attempts. Flies were briefly cold-anesthetized and immobilized using wax. The composition of the internal pipette solution for current clamp recordings in PNs was (in mM): potassium aspartate 140, HEPES 10, MgATP 4, $Na_3$GFP 0.5, EGTA 1, KCl 1, and biocytin hydrazide 13. The internal solution was adjusted to a pH of 7.3 with KOH or aspartic acid and an osmolarity between 262 and 268 mOsm. For voltage-clamp recordings, an equal concentration of cesium was substituted for potassium. The external solution was *Drosophila* saline containing (in mM): NaCl 103, KCl 3, N-tris(hydroxymethyl) methyl-2-aminoethanesulfonic acid 5, trehalose 8, glucose 10, $NaHCO_3$ 26, $NaH_2PO_4$ 1, $CaCl_2$ 1.5, and $MgCl_2$ 4. The pH of the external solution was adjusted to 7.2 with HCl or NaOH (when bubbled with 95% $O_2$/5% $CO_2$), and the osmolarity was adjusted to ~270–275 mOsm. Recording pipettes were fabricated from borosilicate glass and had a resistance of ~6–8 MΩ. Recordings were acquired with a MultiClamp 700B (Axon Instruments) using a CV-7B headstage (500 MΩ). Data were low-pass filtered at 5 kHz, digitized at 10 kHz, and acquired in MATLAB (Mathworks, Natick, MA, USA). Voltages are uncorrected for liquid junction potential.

Flies were removed from the odor exposure environment at least 1 hr prior to recordings, briefly cold-anesthetized anesthetized, and recordings were performed at room temperature. Recordings with respect to experimental groups (odor-exposed versus solvent-exposed) were conducted in a quasi-randomized order; the experimenter was not masked to experimental condition. PN somata were visualized with side illumination form an infrared LED (Smartvision) under a 40× water immersion objective on an upright compound microscope equipped with an epifluorescence module and 488 nm blue light source (Sutter Lambda TLED+). Whole-cell patch-clamp recordings were targeted to specific PNs types based on GFP fluorescence directed by genetic drivers. Three PN types were targeted in this study: DL5 and VM7 using *NP3481-Gal4* and VA6 using *MZ612-Gal4*. One neuron was recorded per brain. PN identity was confirmed using diagnostic odors, and the morphology (and thus identity) of all PNs was further verified posthoc by streptavidin staining in fixed brains. Cells with little or no spontaneous activity upon break-in, suggesting antennal nerve damage, were discarded. Input resistance was monitored on every trial in real-time, and recordings were terminated if the input resistance of the cell drifted by more than 20%.

For experiments in *Figure 7*, the antennal nerves were bilaterally severed immediately prior to recordings using fine forceps.

## ORN recordings

Single-sensillum recordings were performed essentially as previously described (*Bhandawat et al., 2007*). Flies were removed from the odor exposure environment at least 1 hr prior to recordings and briefly cold-anesthetized, and recordings were performed at room temperature. Briefly, flies were immobilized in the end of a trimmed pipette tip using wax, and one antenna or one palp was visualized under a 50× air objective. The antenna was stabilized by tightly sandwiching it between a set of two fine glass hooks, fashioned by gently heating pipettes pulled from glass capillaries (World Precision Instruments, TW150F-3). The palp was stabilized from above with a fine glass pipette pressing it firmly against a glass coverslip provided from below. A reference electrode filled with external saline (see above) was inserted into the eye, and a sharp saline-filled glass recording microelectrode was inserted into the base of the selected sensillum under visual control. Recordings from ab4 and pb1 sensilla were established in the antenna and palp, respectively, based on the characteristic size and morphology of the sensillum, its position on the antenna, and the presence of two distinct spike waveforms, each having a characteristic odor sensitivity and spontaneous firing frequency (*de Bruyne et al., 2001*). Signals were acquired with a MultiClamp 700B amplifier, low-pass filtered at 2 kHz, digitized at 10 kHz, and acquired in MATLAB. Single-sensillum recordings were performed in 2 day old *NP3481-Gal4, UAS-CD8:GFP* females to allow direct comparisons to PN data.

## Odor stimuli

Odors used in this study were benzaldehyde, 2-butanone, *p*-cresol, geosmin, geranyl acetate, 2-heptanone, E2-hexenal, isobutyl acetate, and pentyl acetate. All odors were obtained from MilliporeSigma or Fisher Scientific at the highest purity available (typically >99%). Odor stimuli are referred to by their v/v dilution factor in paraffin oil. Each 20 ml odor vial contained 2 ml of diluted odor in paraffin oil. Diagnostic stimuli that distinguished targeted PN types from other labeled PN types in the driver line used were as follows: for DL5 PNs, E2-hexenal ($10^{-8}$) and benzaldehyde ($10^{-4}$); for VM7 PNs, 2-butanone ($10^{-7}$) and isobutyl acetate ($10^{-4}$); and for VA6 PNs, geranyl acetate ($10^{-6}$). Diagnostic stimuli for ORN classes were as follows: for the ab4 sensillum on the antenna, E2-hexenal ($10^{-7}$) for the ab4a "A" spike and geosmin ($10^{-2}$) for the ab4b "B" spike; and for the pb1 sensillum on the palp, 2-butanone ($10^{-5}$) for the pb1a "A" spike and *p*-cresol ($10^{-3}$) for the pb1b "B" spike.

Fresh odor dilutions were made every 5 days. Each measurement in a fly represents the mean of five trials for ORN responses or six trials for PN responses, spaced 40 s apart. Solvent (paraffin oil) trials were routinely interleaved to assess for contamination. Stimuli were presented in pseudo-randomized order, except for measurements of concentration-response curves where odors were presented from low to high concentrations.

Odors were presented during recordings from olfactory neurons essentially as previously described (*Bhandawat et al., 2007*). In brief, a constant stream of charcoal-filtered air (2.22 L/min) was directed at the fly, with a small portion of the stream (220 ml/min) passing through the headspace of a control vial filled with paraffin oil (solvent) prior to joining the carrier stream (2.0 l/min). Airflow was controlled

using mass flow controllers (Alicat Scientific). When triggered by an external voltage command, a three-way solenoid valve redirected the small portion of the stream (220 ml/min) from the solvent vial through the headspace of a vial containing odor for 500 ms; thus, the concentration of the odor in gas phase was further diluted ~10fold prior to final delivery to the animal. The solvent vial and the odor vial entered the carrier stream at the same point, ~10 cm from the end of the tube. The tube opening measured ~4 mm in diameter and was positioned ~1 cm away from the fly. We presented 500 ms pulses of odor unless otherwise indicated.

Odor mixtures (*Figure 4*) were generated by mixing in air. In these experiments, a second solenoid valve was added that diverted another small fraction of the carrier stream (220 ml/min) through either a second solvent vial or a second odor vial before rejoining the carrier stream. When the two solenoids were both triggered, they drew from the carrier stream at the same point, and the two odorized streams also both rejoined the carrier stream at about the same point, ~10 cm from the end of the delivery tube.

## Immunohistochemistry

Intracellular biocytin fills were processed as previously described (*Wilson et al., 2004*). In brief, brains were fixed for 14 min at room temperature in freshly prepared 4% paraformaldehyde, incubated overnight in mouse nc82 primary antibody (1:40, Developmental Studies Hybridoma Bank #AB_2314866), then subsequently incubated overnight in Alexa Fluor 568 streptavidin conjugate (1:1000, Molecular Probes) and Alexa Fluor 633 goat anti-mouse (1:500, Molecular Probes). PN morphologies were reconstructed from serial confocal images through the brain at 40× magnification and 1 μm step size.

LN innervation was quantified in flies of genotype *+/UAS-brp.S-mStraw; 20XUAS-IVS-CD8:GFP/ NP3056-Gal4*; the *brp.S-mStraw* signal was not measured. Immediately after dissection, brains were fixed for 14 min in freshly prepared 4% paraformaldehyde and incubated overnight in rat anti-CD8 (1:50, Thermo Fisher #MA5-17594) and mouse nc82 (1:40) primary antibodies, then subsequently incubated overnight in Alexa Fluor 488 goat anti-rat (1:500, Abcam #ab150157) and Alexa Fluor 633 goat anti-mouse (1:500, Thermo Fisher #A21050) secondary antibodies. All steps were performed at room temperature, and brains were mounted and imaged in Vectashield mounting medium (Vector labs). In pilot experiments, we compared direct GFP fluorescence in lightly fixed brain with amplified GFP signal using the standard protocol and observed weaker but qualitatively similar signals. Confocal z-stacks at 1024×1024 resolution spanning the entire volume of the antennal lobe were collected on a Leica SP8 confocal microscope at 1 μm slice intervals using a 63×oil-immersion lens. Identical laser power and imaging settings were used for all experiments.

## Calcium imaging

Calcium imaging of ab4a ORN terminals in the DL5 glomerulus was performed on 2-day-old female flies essentially as previously described (*Hong and Wilson, 2015*). In brief, flies were cold-anesthetized and immobilized with wax. The antennal lobes were exposed by removal of the dorsal flap of head cuticle, and the brain was perfused with *Drosophila* saline (see above) that was cooled to 21°C (TC-324C, Warner Instruments) and circulated at a rate of 2–3 ml/min. GCaMP6f signals were measured on a two-photon microscope (Thorlabs, Sterling, VA, USA) using a Ti-Sapphire femtosecond laser (MaiTai eHP DS, Spectra-Physics) at an excitation wavelength of 925 nm, steered by a galvo-galvo scanner. Images were acquired with a 20× water-immersion objective (XLUMPLFLN, Olympus) at 256×96 pixels, a frame rate of 11 Hz, and a dwell time of 2 μs/pixel. The same laser power and imaging settings were maintained for all experiments. The microscope and data acquisition were controlled using ThorImage 3.0. The DL5 glomeruli were clearly labeled as bilateral spherical structures ~10 μm from the dorsal surface of the antennal lobes. In each trial, an 8 s period of baseline activity was collected immediately prior to stimulus presentation which was used to establish the level of baseline fluorescence of each pixel. Each odor stimulus was presented for 500 ms, for three trials, with a 45 s interstimulus interval.

## Optogenetic stimulation of ORN axons

A stock solution of ATR (Sigma-Aldrich, R2500) was prepared at 35 mM in 95% ethanol and stored at –20°C in the dark. The cross that generated the experimental flies was maintained in the dark. Newly eclosed experimental flies for optogenetic experiments were transferred to standard cornmeal/

molasses food supplemented with 350 µM ATR mixed into the food and exposed to odor (or solvent) for 2 days in the dark.

After exposing the antennal lobes, the antennal nerves were acutely and bilaterally severed at their distal entry point into the first segment of the antennal, eliminating EPSCs derived from spontaneous ORN spiking. Electrophysiology rigs were light-proofed. Whole-cell recordings in voltage-clamp mode were established from DL5 PNs in flies expressing Chrimson in all ab4a ORNs (from *Or7a-lexA*). DL5 PNs were identified based on GFP expression (from *NP3481-Gal4*) and the presence of light-evoked responses, and their identity was confirmed after the recording by processing the biocytin fill. In pilot experiments, we tested several methods for optical stimulation and were unable to achieve reliable stimulation of only a single ORN axon presynaptic to the recorded PN using excitation with 590 nm light from a fiber optic-coupled LED (Thorlabs M590F3 with Ø200 µm fiber, 0.22 NA). Excitation of ORN terminals was very sensitive to the position and angle of the optrode relative to the antennal nerve, and we found large variability in the amount of light (intensity and pulse duration) required to evoke EPSCs in the PN.

As an alternative approach, we used wide-field illumination from a 470 nm (blue) LED light source (Sutter Instrument, TLED+). We delivered light pulses of 100 µs duration at 30 s trial intervals, starting at very low levels of light (<0.1 mW/mm$^2$) and gradually increasing the light intensity until an EPSC was observed in an all-or-none manner. At the threshold intensity, trials that fail to evoke an EPSC were infrequently interleaved with successful trials. The ORN-PN synapse is highly reliable, with a single ORN spike evoking robust release of many synaptic vesicles at the ORN terminal (*Kazama and Wilson, 2008*); thus, we interpret these failures as a failure to recruit a spike in a presynaptic axon on that trial (as opposed to a failure in synaptic transmission). As the light intensity was further increased, EPSC amplitude remained relatively constant until it was observed to suddenly double, reflecting the recruitment of a second ORN axon. The mean uEPSC for each PN was determined by averaging the evoked EPSC in ~8–12 trials at a light intensity approximately halfway between the initial threshold intensity and the doubling intensity. Recordings were discarded if any of the following criteria occurred: (1) a high rate of failures at the light intensity chosen for data collection (>10%); (2) uEPSC amplitude was not stable over a range of at least ~5 µW spanning the chosen light intensity; (2) the shape of the uEPSC was not stable.

## Data analysis

Unless otherwise stated, all analyses were performed in MATLAB (Mathworks, Natick, MA, USA).

### Quantification of electrophysiological responses

Analysis of neural responses was performed masked to the experimental condition (odor- versus solvent-exposed) of the recording. For each odor stimulus measurement in a fly, a trial block was composed of five stimulus presentations for ORN responses or six presentations for PN responses, at an intertrial interval of 40 s. The first trial for PN responses was not included in the analysis. Spike times were determined from raw ORN and PN voltage traces using custom scripts in MATLAB (source code available at https://doi.org/10.5061/dryad.v15dv420q) that identified spikes by thresholding on the first and second derivatives of the voltage. All spikes were manually inspected. Spike times were converted into a peristimulus time histogram (PSTH) by counting the number of spikes in 50 ms bins, overlapping by 25 ms. Single-trial PSTHs were averaged to generate a mean PSTH that describes the odor response for each cell. For membrane potential, single-trial voltage traces were averaged to generate a mean depolarization response for each cell. For each DL5 and VM7 PN, the response magnitude for each stimulus was computed as the trial-averaged spike rate (or membrane potential) during the 500 ms odor stimulus period, minus the trial-averaged spontaneous firing rate (or membrane potential) during the preceding 500 ms. For VA6 PNs, the response magnitude was computed over a 1000 ms window that begins at stimulus onset, which better captured the protracted odor response (which extends into the post-stimulus period) observed in this cell type. A mean response magnitude was computed across trials for each experiment, and the overall mean response was plotted as mean ± SEM across all experiments in each condition.

## Analysis of LN anatomical innervation

Images of all antennal lobes were collected with identical laser power and imaging settings (magnification, detector gain, offset, pixel size, and dwell time). Brains in which >0.01% of pixels in neuropil regions (excluding cell bodies and primary neurites) were high or low saturated were rejected. Confocal image stacks were imported into ImageJ (NIH) for analysis. Analysis of LN innervation was conducted masked to the experimental condition (odor- versus solvent-exposed). The boundaries of glomeruli of interest were manually traced in every third slice using the nc82 neuropil signal, guided by published atlases (*Couto et al., 2005*; *Laissue et al., 1999*) and then interpolated through the stack to obtain the boundaries in adjacent slices. The 3D region of interest (ROI) manager plugin (*Ollion et al., 2013*) was used to group together sets of ROIs across slices corresponding to each glomerulus to define the volumetric boundaries for each glomerulus and was then used for quantification of glomerular volume (number of pixels) and pixel intensities in each channel. For each 3D ROI corresponding to an individual identified glomerulus, LN neurites or neuropil per volume were computed as the sum of the pixel values in the ROI in the anti-CD8 (LN neurites) or nc82 (neuropil) channels, respectively, divided by the total number of pixels. The ratio of LN neurites to neuropil was computed as the sum of the pixel values in the ROI in the anti-CD8 channel divided by the sum of the pixel values in the ROI in the nc82 channel. To combine measurements across all glomeruli for a given metric (e.g. volume, LN neurites per volume, etc.), the measurement for each glomerulus in an experiment was normalized to the mean value for that glomerulus across all experiments in the solvent-exposed control condition. Plots show mean and SE across brains.

## Analysis of calcium imaging

The duration of each calcium imaging trial was 15 s, collected at 11 frames per second and 256×96 pixels per frame. Stimulus-evoked calcium signals (ΔF/F) were quantified from background-subtracted movies as the change in fluorescence ($F-F_0$) normalized to the mean fluorescence during the baseline period of each trial ($F_0$, averaged over 70 frames immediately preceding the odor), computed on a pixel-by-pixel basis in each frame. A Gaussian lowpass filter of size 4×4 pixels was applied to raw ΔF/F heatmaps.

A ROI was manually traced around each DL5 glomerulus, which contain the ab4a ORN terminals. ΔF/F signals were averaged across the pixels in the two DL5 ROIs and across three trials for each stimulus in an experiment. The odor response to the 500 ms stimulus presentation was typically captured in ~6 frames. The peak response for each experiment was quantified from the frame containing the maximum mean ΔF/F signal during the stimulus presentation. The overall mean response was plotted as mean ± SEM across all experiments in each condition.

## Statistics

A permutation analysis was used to evaluate differences between experimental groups because of its conceptual simplicity and because it does not require assumptions about the underlying distribution of the population. For each measurement (e.g. the response of a PN type to an odor stimulus), experimental observations from flies in odor- and solvent-exposed conditions were combined and randomly reassigned into two groups (maintaining the number of samples in each respective experimental group), and the difference between the means of the groups was computed. This permutation process was repeated 10,000 times without replacement to generate a distribution for the difference between the means of the odor- and solvent-exposed groups (see *Figure 6—figure supplement 1* for an example) under the null assumption that there is no difference between the two populations. We calculated the fraction of the empirically resampled distribution which had an absolute value that equaled or exceeded the absolute observed difference between the means of the odor- and solvent-exposed groups to determine the two-tailed p-value that the observed outcome occurred by chance if the populations are not different. The cut-off for statistical significance of α=0.05 was adjusted to account for multiple comparisons in an experiment using a Bonferroni correction. Permutation testing was used in all figures, except in *Figure 5* where the Mann-Whitney *U*-test was used, implemented in MATLAB (Wilcoxon rank sum test).

Sample sizes were not predetermined using a power analysis. We used sample sizes comparable to those used in similar types of studies (e.g. *Das et al., 2011*; *Sachse et al., 2007*; *Bhandawat et al., 2007*). The experimenter was not masked to experimental condition or genotype during data

collection. For a subset of analyses (analysis of electrophysiological data and quantification of LN innervation), the analyst was masked to the experimental condition, as described above.

## Acknowledgements

We thank Chris Potter for the gift of Or7a-KI flies. We thank Meike Lobb-Rabe for generating the *Or7a-lexA* flies. We thank Daniel Wagenaar and the Caltech Neurotechnology Center for technical support in constructing fly holders. We thank members of the Hong laboratory for their critical reading and feedback on this manuscript. This work was supported by grants to EJH from the National Science Foundation (Award #1556230), the Shurl and Kay Curci Foundation, and the National Institutes of Health (1RF1MH117825). EJH was supported by a Clare Boothe Luce professorship, and EM was supported by the Amgen Scholars Program.

## Additional information

### Funding

| Funder | Grant reference number | Author |
| --- | --- | --- |
| National Science Foundation | Award #1556230 | Elizabeth J Hong |
| National Institutes of Health | 1RF1MH117825 | Elizabeth J Hong |
| Shurl and Kay Curci Foundation | | Elizabeth J Hong |
| Clare Boothe Luce professorship | | Elizabeth J Hong |
| Amgen Scholars Program | | Elizabeth G Maurais |

The funders had no role in study design, data collection and interpretation, or the decision to submit the work for publication.

### Author contributions

Zhannetta V Gugel, Conceptualization, Formal analysis, Validation, Investigation, Visualization, Methodology, Writing - original draft, Writing - review and editing; Elizabeth G Maurais, Formal analysis, Investigation, Visualization; Elizabeth J Hong, Conceptualization, Formal analysis, Supervision, Funding acquisition, Methodology, Project administration, Writing - review and editing

### Author ORCIDs

Elizabeth G Maurais ⓘ http://orcid.org/0000-0001-8612-5741
Elizabeth J Hong ⓘ http://orcid.org/0000-0003-3866-418X

### Decision letter and Author response

Decision letter https://doi.org/10.7554/eLife.85443.sa1
Author response https://doi.org/10.7554/eLife.85443.sa2

## Additional files

### Supplementary files

• Supplementary file 1. Complete genotypes and *n* for all experiments. Where only a subset of stimuli is plotted due to space constraints, bolded entries correspond to the stimuli represented in the figure. Solvent is paraffin oil.

• MDAR checklist

### Data availability

The new transgenic fly generated in this study, Or7a-lexA (III), will be deposited in the Bloomington *Drosophila* Resource Center for public distribution. Source data from electrophysiology, functional

imaging, and confocal imaging experiments used to generate Figures 2-7 are publicly available on Dryad repository at https://doi.org/10.5061/dryad.v15dv420q.

The following dataset was generated:

| Author(s) | Year | Dataset title | Dataset URL | Database and Identifier |
|---|---|---|---|---|
| Hong EJ | 2023 | Gugel, Maurais, Hong (2022). Source data figures 2-7 | https://dx.doi.org/10.5061/dryad.v15dv420q | Dryad Digital Repository, 10.5061/dryad.v15dv420q |

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
