## [Editor Report]

Does exposure to odors induce some type of adaptive plasticity at the early layers of the olfactory circuit? In this valuable work, the authors use naturalistic concentrations of odor to provide convincing evidence that there is actually very little plasticity in the projection neurons at the glomerular layer of the system. Their findings show small increases in responses to low concentrations of the exposed odor.

---

## [Decision Letter]

**Decision letter after peer review:**

[Editors’ note: the authors submitted for reconsideration following the decision after peer review. What follows is the decision letter after the first round of review.]

Thank you for submitting the paper "Chronic exposure to odors at naturally occurring concentrations triggers limited plasticity in early stages of *Drosophila* olfactory processing" for consideration by eLife. Your article has been reviewed by 3 peer reviewers, and the evaluation has been overseen by a Reviewing Editor and a Senior Editor. The reviewers have opted to remain anonymous.

Comments to the Authors:

We are sorry to say that, after consultation with the reviewers, we have decided that this work will not be considered further for publication by eLife in its current form. However, should you be able to address the issues raised with additional experimental and textual revisions, we would be happy to reconsider the manuscript.

As you will see in the reviews below, the reviewers all believed that the data were high quality, but there were strong concerns about several aspects of the study, which came out clearly in the discussion and which I elaborate on below.

1) There was a broad question about whether the prior reported results in the field can be reproduced in your lab with these procedures. If not, this null result becomes much harder to interpret as specific to this set of low concentrations, as the authors interpret it here. A positive result seems to be required to interpret the negative one.

(1.1) Related to this question, and assuming prior results are reproducible, is it the pulsing vs. chronic or the concentration difference that results in differences from prior experiments? Since two dimensions are changed (at least) from prior protocols, it seems critical to assess which of these is the important driver of the observed differences.

2) It is unclear whether, at the low concentrations and/or with this pulsed protocol, there is any reason to expect plasticity in these experiments. For instance one reviewer asked about whether any behavioral changes suggest there should be plasticity somewhere in the system under these conditions. The significance of "decreasing concentration lowers modest plasticity to almost undetectable plasticity" seemed to be less than what some reviewers were hoping for.

There was some disagreement on this point among reviewers during the discussion, but this was the prevailing sense.

3) The reviewers agreed that the unexplained variability with odorants and glomeruli (Reviewer 3, point 7) were important to understand better.

4) Given the null result, there was a question about how the authors could verify that the experiment was actually delivering the odor they intended, given issues of PID sensitivity to such low concentrations.

*Reviewer #1 (Recommendations for the authors):*

This paper has a nice logical progression to the experiments, and very well thought-out data presentation and graphics.

Strengths include:

– Odors are used at naturalistic concentrations and there is validation of the PN firing rates elicited in these conditions.

– The interpretation of the results is measured and appropriate. The effects that are observed are relatively small. Rather than focus on these small changes, the overall interpretation is that basically this part of the circuit doesn't change much with olfactory experience, which I think faithfully conveys the reality of the data.

– Technically the data looks very sound, e.g. in cases where there is no effect, the PSTHs for control and manipulated conditions often overly completely throughout the timecourse i.e. responses are extremely consistent.

Weaknesses include:

It would have been nice to have some behavioral experiments to e.g. establish whether the small increase they observe in PN activity reflects increased behavioral sensitivity to the odors.

Not a weakness, but the study does raise a number of questions that stay unresolved:

– If the excitatory network is extremely broad, why is it that activating it via one glomerulus exerts the effect while another glomerulus doesn't?

– Along similar lines, in Fig7 the authors show an effect of chronic exposure on excitatory lateral inputs, but this is only observed with some odors. It is unclear why. Presumably this is a clue to where exactly the plasticity is operating in the network.

Although these questions are not resolved by this study, the overall claims are justified. Of course with a primarily negative result, the question is whether other types of odor exposure protocols might evoke plasticity. However within the well-justified bounds of their experimental design, the authors conclusions are well-supported. And the evidence looks very robust i.e. it appears the results would be highly repeatable.

*Reviewer #2 (Recommendations for the authors):*

Gugel et al. set out to investigate the effects of chronic odor exposure onto the activity of secondary neurons in the *Drosophila* olfactory system. To do so, they established an odor delivery system to administer flies a short puff of odor (1 s) repeatedly (every 20 s) over the course of 2 days. Neural activity is measured from identified olfactory neurons on day 3. Compared to previous studies on chronic exposure, the authors here select odor stimuli that are more likely to be encountered in nature, therefore in the low concentration range, which however still activate reliably the neurons investigated. Altogether, the approach, experiments and analysis are rigorous. Unfortunately and to some degree surprisingly, the chronic exposure has very mild effects on the physiological properties and odor response of olfactory projection neurons (PNs). When present, some plasticity was observed broadly across different PNs, rather than only in those directly activated by the odor, and it consisted in an increased response to low concentrations. These results contrast with previous reports of plasticity induced by chronic stimulation in the same brain area, a discrepancy that can be possibly explained by the different stimulation protocol. The authors therefore conclude that chronic exposure to ecologically relevant stimuli has limited effects on odor encoding. However, it remains unclear whether the stimulation protocol adopted effectively drives sufficient olfactory inputs in the rearing chamber and, consequently, a change in behavior, or if on the contrary is just simply a very weak stimulus which has no physiological and functional consequences.

I find the paper very well presented, the experiments well described and the results well discussed. In my opinion, however, the main question still remains open: is there really no plasticity in the antennal lobe upon chronic stimulation or is the stimulation not sufficient to drive plasticity? how should the results be reconciled with previous studies? More specific points follow below.

1. Did the author try to induce stronger plasticity by using higher concentrations of the odors or continuous delivery? I completely understand the argument for low, natural concentrations, but if one could recapitulate previous results by stronger activation, then these new data could be additively and complementarily interpreted together with previous findings.

2. Another point I would like to be clarified is whether the PID signal measured in the rearing bottle (fig.1B) is similar to the PID signal measured on the ephys rig. I guess that this is unlikely due to the fact that the PID is not sensitive to the low concentrations used in the ephys. Could it be that the low concentrations deplete faster? Or that the odor stimulus in the rearing bottle does not scale linearly with concentration and therefore no stimulus is actually going through?

3. More in general, I think that it is necessary to demonstrate that the chronic stimulation used in this paper induces some kind of change in sensory perception, for example at a behavioral level. If this is not the case, then one could wonder why to embark in such tedious experiments. If there is instead a behavioral change, then one would logically look for physiological correlates in the AL first and secondly in downstream brain areas.

4. The authors end the paper with a discussion about the advantages of not having plasticity. I think that this discussion is a bit biased to justify the current data and does not consider general computational ideas. I would argue that one shouldn't expect plasticity with respect to rather "common" stimuli. The developmental program has evolved to build a sensory system for the "common" environment, plasticity should be a mechanism to cope with variations from the "common". Within this discussion the authors should try clarify what was their initial hypothesis. Why did they expect plasticity in the AL? What did they think this could be useful for?

5. If the plasticity is broad across PNs, and therefore it is not stimulus specific, how can plasticity depend of valence and context, which require stimulus specific information? (refers to discussion ~L629)

*Reviewer #3 (Recommendations for the authors):*

With electrophysiological and optogenetic experiments performed in *Drosophila*, the authors tested the hypothesis that olfactory codes can adapt to the statistical frequency with which specific odors are encountered in natural environments. Focusing on a small number of glomeruli and their cognate odorants, the authors determined, consistent with prior work, that plasticity in this context is subtle, affects mainly local neurons, with exceptions that reveal no new general plasticity rules. The results suggest that chronic odor exposure increases the overall strength of global excitatory coupling among glomeruli in the antennal lobe after chronic odor exposure, possibly heightening the sensitivity of PNs to weak odors.

Strengths of this work are that the research question is interesting, the preparation is appropriate and well-used, experiments appear to have been done well, and the authors usefully report what could be described as negative results (more authors should do this!).

My main concern is that this work will have limited impact. Similar papers have appeared over the years (many are appropriately cited and described in this manuscript). Each such project featured somewhat different choices of stimuli and protocols, and each arrived at somewhat different conclusions. The authors attribute this variance to "methodological differences" (lines 428ff). The present work contributes a useful new dataset to this field. But it does not directly address or resolve the causes of these different results or yield a new unifying understanding of this form of plasticity.

With electrophysiological and optogenetic experiments performed in *Drosophila*, the authors tested the hypothesis that olfactory codes can adapt to the statistical frequency with which specific odors are encountered in natural environments. Strengths of this work are that the research question is interesting, the preparation is appropriate and well-used, experiments appear to have been done well, and the authors usefully report what could be described as negative results (more authors should do this!).

My main concern is that this work will have limited impact. A number of similar papers have appeared over the years (many are appropriately cited and described in this manuscript). Each such project featured somewhat different choices of stimuli and protocols, and each arrived at somewhat different conclusions. The authors attribute this to "methodological differences" (lines 428ff). The present work contributes a useful new dataset to this field. But because it does not directly address or resolve the causes of these different results or yield a new unifying understanding of this form of plasticity, I believe only specialists will want to read about it.

Suggestions for improving the text:

1) line 127ff: "Photoionization measurements of the odor stimulus in the rearing bottle demonstrated that the stimulus was stable across more than 24 hours (Figure 1B)." I disagree with this characterization. To me, the PID signals shown in Fig 1B clearly indicates that the amount of released material systematically and substantially decreases over time. The authors can question whether this amount of change is meaningful, but they should not deny the change.

2) line 132: "with little adaptation of the PN response to the odor..." Similarly, Fig 1D also shows systematic decreases over trials and very dramatic adaptation during the course of each response.

3) Fig 1C: the authors do not note, but should, that the response duration increases systematically and dramatically over trials.

4) line 133ff: "odor stimuli to which we exposed flies were of significantly lower concentration than what has been used in prior studies investigating olfactory plasticity...." This statement calls for more explanation and citations of these prior studies.

5) line 177f: "Due to the small size of VM7 PN somata, VM7 PN spikes are small..." I'm not sure I follow this logic: many especially small insect neurons generate especially big spikes, and some big neurons generate small spikelets.

6) line 233f, 405ff, and elsewhere: "This result implicates local lateral circuitry in the antennal lobe in odor-experience dependent neural plasticity." Stopfer and Laurent 1999 (not cited) found similar results in locust exposed to a relatively brief series of odor pulses suggesting many hours of exposure may not be necessary to evoke this plasticity. Subsequent modeling work (Bazhenov et al, 2005, not cited) showed these findings can be explained by activity-dependent facilitation of AL inhibitory synapses, and lead to more reliable odor responses. The authors may wish to consider this work.

7) lines 259f: "Unexpectedly, in flies chronically exposed to E2-hexenal only, many glomeruli were smaller in volume than their counterparts in solvent-exposed flies." This is a potentially worrisome indication that the flies may have been adversely affected in a general way by the chronic exposure. Did the exposed flies appear healthy compared to controls? Can the authors rule out that they had become sick?

---

## [Author Response]

[Editors’ note: the authors resubmitted a revised version of the paper for consideration. What follows is the authors’ response to the first round of review.]

As you will see in the reviews below, the reviewers all believed that the data were high quality, but there were strong concerns about several aspects of the study, which came out clearly in the discussion and which I elaborate on below.

We appreciate the opportunity to submit a revised manuscript. We believe new data and significant textual revisions, especially in the Introduction & Discussion, address the reviewers’ major concerns, and more directly communicate the significance of this work for the field of experience-dependent sensory plasticity.

1) There was a broad question about whether the prior reported results in the field can be reproduced in your lab with these procedures. If not, this null result becomes much harder to interpret as specific to this set of low concentrations, as the authors interpret it here. A positive result seems to be required to interpret the negative one.(1.1) Related to this question, and assuming prior results are reproducible, is it the pulsing vs. chronic or the concentration difference that results in differences from prior experiments? Since two dimensions are changed (at least) from prior protocols, it seems critical to assess which of these is the important driver of the observed differences.

The reviewers identify an important point regarding whether past results on the effects of odor exposure at high concentration on PN odor coding can be reproduced in our lab. When we initiated the study, we originally took the perspective that, if the effects of chronic odor exposure on olfactory function were mediated solely through increases in neural activity in ORNs (as is the common assumption), then odor stimuli at concentrations that are high enough to excite ORNs to their maximal rates of firing would be appropriate to study the impact of increased olfactory experience on the system. Increasing the odor concentration does not further increase levels of odor-evoked firing in the corresponding ORN class (see Figure 2) and would only act to excite additional classes of ORNs (which complicates the evaluation of the specificity of plasticity).

However, we take the reviewers’ concerns seriously and acknowledge the usefulness of evaluating the role of odor concentration in these experiments to enable a clearer interpretation of the results in this study. Admittedly, every study in this field in the past has used very high concentrations of odor for chronic exposure. However, we should note it may be impossible for us to fully deduce what happened in another person’s laboratory a decade or more ago, if only because technical constraints prevent us from exactly replicating their methodologies (see below). Thus, we provide an experiment that tests the basic question asked by reviewers: does chronic sustained (non-pulsed) exposure to a monomolecular odor at high concentration elicit a decrease in PN sensitivity, as previously reported (Sachse et al. 2007, Das et al. 2011)?

These two prior studies chronically exposed flies to either 5% CO_2_ or a 20% ethyl butyrate (EB) source. The CO_2_-sensitive V PNs are ventrally located in the antennal lobe and are not readily accessible for visually targeted electrophysiological recordings in our preparation, complicating the use of CO_2_ exposure for this experiment. We also considered EB: using calcium imaging in ORN terminals in the antennal lobe, we’ve observed that 20% EB activates a large number of ORN classes (~35-40% of glomeruli), including the DM5 glomerulus previously studied in Das et al. 2011 (identified by position and sensitivity to 0.001% ethyl 3-hydroxybutyrate). Like in Das et al. 2011, we placed a glass vial with a perforated cap containing 1 ml of 20% EB in paraffin oil in the growth bottle of newly eclosed flies in which the DM5 PN is fluorescently labeled (T2-Gal4, UASCD8GFP); we replaced the odor source daily. After four days, more than half of the flies were dead and the remaining flies were lethargic and did not appear qualitatively normal, with significant debris coating their body surface, which is not typically observed. We attempted the experiment twice, using a newly purchased bottle of ethyl butyrate the second time, and observed the same outcome both times. We can only speculate that the difference between how 20% EB affects our flies versus those in Das et al. 2011 may be due to differences in genetic background. The flies in Das et al. 2011 show behavioral responses to odor, which our flies would not be able to do because of low walking speed. Due to the poor condition of the flies, we did not proceed with this odor stimulus.

Therefore, we decided to evaluate how flies respond to chronic exposure to 20% geranyl acetate (GA). We made this decision for several reasons: (1) even at this high concentration (>1000x higher than is used in Figure 2), geranyl acetate still preferentially activates the ab5a ORNs that project to glomerulus VA6, allowing us to evaluate the effect on PNs receiving direct versus indirect chronic input; and (2) we already had previous data for exposure to pulsed 0.01% GA, so a comparison of pulsed vs. sustained exposure in this glomerulus was possible. We followed the odor exposure method used in previous studies, placing the 20% GA odor source in the growth bottle of the fly, and exposed flies to odor for four days. We recorded from VA6 PNs and DM6 PNs in 5-day old flies, a very technically challenging experiment because patch-clamp recordings become significantly more difficult in older animals as the glial sheath surrounding the neurons becomes stickier. The results of this experiment are presented in the new Figure 4 and accompanying text.

We found that chronic, sustained exposure to 20% GA for four days (Figure 4) had qualitatively similar effects on PN odor coding as was observed after chronic, intermittent exposure to 0.01% GA (Figure 2K-P). Sustained exposure to high concentrations of GA elicited either no change or moderate increases in PN responses to odor. As before, PN plasticity was more consistent for weaker odor stimuli, and the increased responsiveness of PNs to odor was observed in both VA6 and DM6 PNs, so plasticity was not selective for PNs corresponding to the glomerulus receiving chronic direct excitation. These data show that sustained odor exposure to a high concentration of geranyl acetate for four days does not elicit reduced PN responses.

Our data are overall most consistent with a scenario where chronic excitation of the olfactory system elicits limited plasticity in PNs, increasing the sensitivity of PNs to weak odor stimuli that activate PNs towards the bottom of their dynamic range. In general, PN responses to moderate or strong odor stimuli are stable and unaffected by chronic odor exposure. Reduced PN activity was not observed in flies reared in any of the four odor exposure environments we studied. We note that a more recent study from Kidd et al., 2015 using calcium imaging found that chronic exposure to 1% geranyl acetate mildly enhanced VA6 PN responses to this odor, so we don’t think the difference between our results and Sachse et al. 2007 and Das et al. 2011 is explained by recording methodology. Additionally, a recent, similarly designed study that imaged mitral cell odor responses in mice chronically exposed to odors also observed increases in odor-evoked mitral cell activity (Liu and Urban, 2017), and this plasticity was observed broadly across many glomeruli, including those not directly chronically stimulated.

We cannot, of course, rule out the possibility that the difference between our results and Kidd et al. 2015 versus Sachse et al. 2007 and Das et al. 2011 is due to differences in the specific odors and glomeruli studied. Methodological differences may also be relevant: more recent studies, including this study, use less invasive, more intact live recording preparations, and also use a newer generation of GCaMP calcium indicators or electrophysiological recordings, which may allow better sampling.

2) It is unclear whether, at the low concentrations and/or with this pulsed protocol, there is any reason to expect plasticity in these experiments. For instance one reviewer asked about whether any behavioral changes suggest there should be plasticity somewhere in the system under these conditions. The significance of "decreasing concentration lowers modest plasticity to almost undetectable plasticity" seemed to be less than what some reviewers were hoping for.There was some disagreement on this point among reviewers during the discussion, but this was the prevailing sense.

If the current accepted model for the circuit mechanisms mediating experience-dependent olfactory behavioral plasticity is true (for instance, see reviews by Bai and Suzuki, 2020; or Golovin and Broadie, 2016), there is good reason to expect to see PN plasticity under our experimental conditions. We recognize that a major concern is that the concentration of odor we used for chronic exposure is much lower than used in prior studies. Despite our comparatively lower odor concentrations, we emphasize that the odor environment elicited high, saturating firing rates in the corresponding ORNs and PNs excited by the odor. That is, using a higher concentration of odor would not reliably drive higher firing rates in the PN type that is directly excited by that stimulus (it would only be expected to start recruiting direct excitation to other glomeruli). Our choice of odor exposure conditions was intended to rigorously test a dominant hypothesis in the field that reduced behavioral aversion to an odor following chronic exposure is mediated by selective strengthening of GABAergic inhibition in glomeruli that are chronically excited by over-represented odors in the environment, but not in glomeruli experiencing typical levels of activity. We believed the most direct way to test this model was to strongly, selectively, and chronically increase direct excitatory input to a single glomerulus and evaluate the impact on the PNs corresponding to that glomerulus.

Furthermore, we have now shown that, for geranyl acetate, sustained exposure to high concentrations of odor for a full four days still does not elicit reduced odor responses in the PN receiving direct chronic excitation. We conclude that the commonly accepted model for experience-dependent olfactory plasticity is, at the very least, not true for all glomeruli. We do not definitely claim that prior studies (Sachse et al; Das et al. 2011) are incorrect because we do not replicate their experiments exactly. However, we are presenting a set of experiments that are premised on the plasticity model that these influential studies introduced into the literature, and our data are inconsistent with the most commonly accepted model. The prevailing view predicts that we should observe decreases in odor-evoked responses in PNs processing overrepresented odors in the environment. In fact, any PN plasticity we (and other recent studies, see above) have observed functions in the opposite direction. We believe the major contribution of our study is that we have provided a high-quality dataset that rigorously tests and calls into question an influential, widely accepted model for olfactory plasticity. Our results indicate we need to revisit the commonly accepted model for PN olfactory plasticity; at a minimum, it does not hold for all glomeruli. In the Discussion section “Implication for understanding odor experience-dependent changes in olfactory behavior,” we discuss the significance of these results for our interpretation of how animals’ behavior towards an odor changes after chronic exposure in early life. Establishing whether PN plasticity occurs robustly, and, if so, in what direction it acts, is critical to determining if changes in olfactory behavior are correctly interpreted as habituation (also, see below in the response to Reviewer 3).

3) The reviewers agreed that the unexplained variability with odorants and glomeruli (Reviewer 3, point 7) were important to understand better.

Flies chronically exposed to E2-hexenal (10^-7^, e.g., 0.00001%) appear qualitatively normal – they walk, climb the sides of the vial, and attempt to escape in a manner indistinguishable from unexposed flies. In contrast, in our hands, exposure to some odors at high concentration (e.g., 20% ethyl butyrate) can cause flies to become unhealthy and reduce their locomotion (see above in point (1)). We do not understand why antennal glomeruli are smaller after exposure to E2-hexenal. We did note that the fact that glomeruli are broadly affected could relate to the broad impact of odor exposure across glomeruli (beyond just the glomerulus receiving chronic direct excitation, e.g. Figure 3). The LN anatomical data has been moved to the supplement, to better focus the paper on the major results on how odor exposure impacts PN odor coding.

4) Given the null result, there was a question about how the authors could verify that the experiment was actually delivering the odor they intended, given issues of PID sensitivity to such low concentrations.

We should clarify that we don’t consider our results a “null” result – we consistently observe mild to moderate increases in PN responses to weak odor stimuli in flies chronically exposed to E2-hexenal (10^-7^) or 2-butanone (10^-4^). This result is seen at the level of membrane depolarization or firing rate in all three PN types (DL5, VM7, VA6) investigated.

It’s not completely clear if the reviewers are referring to odor delivery during chronic exposure or during neural recordings. The PID has sufficient sensitivity to detect 2-butanone (10^-4^) in the odor exposure environment (e.g., in the growth bottle). The degree of stability of a pulse of 2butanone (10^-4^) over 24 hours is shown in Figure 1B; the odor source is refreshed every 12 hours so the amplitude of the odor pulse should be stable to within ~20%. We have no reason to believe that the headspace above other odor solutions should act any differently when moved through the olfactometer. In pilot experiments, we saw that the adaptation of peak PN responses across trials (e.g., Figure 1C), which reflects both neural adaptation and odor source rundown/ depletion, was qualitatively similar for all three odor stimuli. This suggests that these odors are unlikely to be extremely different in terms of their relative rates of equilibration between the liquid odor source and the headspace. If the reviewers are referring to odor delivery during ephys measurements, it’s true that during ORN and PN recordings we used much lower odor concentrations. However, the stimulus-locked response of the neuron indicates the stimulus was being delivered and detected by the fly antennae.

Reviewer #1 (Recommendations for the authors):This paper has a nice logical progression to the experiments, and very well thought-out data presentation and graphics.Strengths include:– Odors are used at naturalistic concentrations and there is validation of the PN firing rates elicited in these conditions.– The interpretation of the results is measured and appropriate. The effects that are observed are relatively small. Rather than focus on these small changes, the overall interpretation is that basically this part of the circuit doesn't change much with olfactory experience, which I think faithfully conveys the reality of the data.– Technically the data looks very sound, e.g. in cases where there is no effect, the PSTHs for control and manipulated conditions often overly completely throughout the timecourse i.e. responses are extremely consistent.

We appreciate the reviewer’s evaluation on the high technical quality of the study.

Weaknesses include:It would have been nice to have some behavioral experiments to e.g. establish whether the small increase they observe in PN activity reflects increased behavioral sensitivity to the odors.

We agree that demonstrating an effect of chronic exposure to low, naturalistic concentrations of monomolecular odor on behavior would be powerful. Though flies can obviously detect such stimuli at the level of neural activity, flies do not behave towards monomolecular odors at these low concentrations in standard lab olfactory assays like a trap assay or a two-choice assay. This does not necessarily indicate the fly cannot perceive the odor, they may just not be motivated to act in response to it in these contexts. For instance, there are also many monomolecular odor stimuli of high concentrations to which flies do not behave.

In other experiments in our lab, we have shown that chronic exposure of flies to natural odor sources (e.g. banana) can elicit long-lasting increases in behavioral attraction to the odor, and monomolecular volatiles occur in such natural sources in a similar range (~1 ppb to 10 ppm in air). However, conducting the experiments in this study using natural sources that are complex mixtures of monomolecular odors would have prevented us from evaluating whether olfactory plasticity acts selectively in the glomerulus receiving chronic direct excitation.

Not a weakness, but the study does raise a number of questions that stay unresolved:– If the excitatory network is extremely broad, why is it that activating it via one glomerulus exerts the effect while another glomerulus doesn't?– Along similar lines, in Fig7 the authors show an effect of chronic exposure on excitatory lateral inputs, but this is only observed with some odors. It is unclear why. Presumably this is a clue to where exactly the plasticity is operating in the network.

These are very good questions. We note that new data that examines the impact of chronic exposure to high concentrations of geranyl acetate (20%) in two glomeruli (the direct PN VA6 and an indirect PN DM6) shows that chronic excitation of VA6 elicits increases in DM6 PN excitability (but VA6 PN responses only trend higher). Thus all three odors we tested, E2hexenal, 2-butanone, and geranyl acetate, consistently elicit mild increases in odor-evoked responses in some PNs. Overall, the largest and most consistent effects of chronic odor exposure are, in fact, observed in PNs receiving indirect input from the chronically activated ORN class, further implicating the lateral networks in the weak olfactory plasticity observed in the antennal lobe.

Although the lateral excitatory and inhibitory networks interconnect glomeruli broadly in an allto-all manner, the absolute magnitude of lateral excitation and inhibition recruited by different odors varies. This is due at least in part to differences in the strength of coupling of different ORN or PN types to inhibitory LNs (iLNs) and excitatory LNs (eLNs), respectively, such that different glomeruli (and thus odors) vary in the efficacy with which they recruit lateral inhibition (Hong and Wilson, 2015) and excitation (Yaksi and Wilson, 2010). Additionally, different PN types vary in their sensitivity to lateral inhibition or lateral excitation recruited by a given odor. We speculate that chronic excitation of different ORN classes may differently engage these lateral networks such that chronic activity in the lateral network may vary for different odors and result in different effects on network plasticity.

In the case of Figure 7H-J, one possibility for why not all test odors elicit increased lateral excitation after chronic exposure to geranyl acetate is that chronic excitation of a specific ORN class does not equivalently strengthen PN-eLN connections in all glomeruli. Another (nonmutually exclusive) possibility is that some test odors elicit levels of PN activity in their target PNs that saturate the PN-eLN connection and do not allow further increases in PN-eLN coupling to be resolved. eLN activity is highly non-linear with respect to olfactory input: eLNs are sensitive to very low levels of olfactory input and saturate quickly as odor concentration increases. 2-butanone and p-cresol, two of the odors that show the least effect, are also two odors in our panel that are more narrowly activating (e.g. exciting fewer PN types strongly, see Hallem and Carlson, 2006) compared to other odors that broadly excite many more PN types, some of which may be excited to lower rates of firing that are within the dynamic range of PNeLN recruitment.

Although these questions are not resolved by this study, the overall claims are justified. Of course with a primarily negative result, the question is whether other types of odor exposure protocols might evoke plasticity. However within the well-justified bounds of their experimental design, the authors conclusions are well-supported. And the evidence looks very robust i.e. it appears the results would be highly repeatable.

We thank the reviewer for noting the degree of sampling and technical quality of our measurements. We agree negative results are always difficult to interpret, which motivated us to collect new data examining the impact of exposure to very high concentrations of monomolecular odor. Although negative results are disappointing, we believe, in this particular case, the publication of stringently peer-reviewed negative results is important because the original results describing PN adaptation to chronic odor exposure have so strongly influenced the interpretation of the behavioral consequences of long-term olfactory experience (see section in Discussion: “Implications for understanding odor experience-dependent changes in olfactory behavior).”

Reviewer #2 (Recommendations for the authors):Gugel et al. set out to investigate the effects of chronic odor exposure onto the activity of secondary neurons in the *Drosophila* olfactory system. To do so, they established an odor delivery system to administer flies a short puff of odor (1 s) repeatedly (every 20 s) over the course of 2 days. Neural activity is measured from identified olfactory neurons on day 3. Compared to previous studies on chronic exposure, the authors here select odor stimuli that are more likely to be encountered in nature, therefore in the low concentration range, which however still activate reliably the neurons investigated. Altogether, the approach, experiments and analysis are rigorous. Unfortunately and to some degree surprisingly, the chronic exposure has very mild effects on the physiological properties and odor response of olfactory projection neurons (PNs). When present, some plasticity was observed broadly across different PNs, rather than only in those directly activated by the odor, and it consisted in an increased response to low concentrations. These results contrast with previous reports of plasticity induced by chronic stimulation in the same brain area, a discrepancy that can be possibly explained by the different stimulation protocol. The authors therefore conclude that chronic exposure to ecologically relevant stimuli has limited effects on odor encoding. However, it remains unclear whether the stimulation protocol adopted effectively drives sufficient olfactory inputs in the rearing chamber and, consequently, a change in behavior, or if on the contrary is just simply a very weak stimulus which has no physiological and functional consequences.

We demonstrate that the odor stimuli to which flies are chronically exposed drive high, near saturating rates of firing in the ORN class they each activate (which presumably expresses the highest affinity receptor for the odor, Figures 1 and 2). If the concentration of the monomolecular odor is further increased, it does not drive consistently higher rates of firing in their target ORNs or PNs (our own observations, and also many other studies in this system, for instance, see Bhandawat et al. 2007; Olsen et al. 2010). Increasing odor concentration would only act to elicit activity in additional classes of ORNs that have lower affinity receptors for the odor. Thus, we believe the odor stimuli we chose for chronic exposure are appropriate for testing the hypothesis that strong, persistent excitation of a given ORN input elicits plasticity in the antennal lobe circuit selectively in the PNs that directly process that input.

However, since reviewers were concerned about the low concentrations of the odors in our exposure protocol, we now present new data from animals exposed to very high concentrations of geranyl acetate (Figure 4). In these new experiments, we observe qualitatively similar results to what was observed in flies reared in our low concentration exposure environments. We are unable to create conditions using our methods that reproduce the original observation of reduced PN sensitivity in flies chronically exposed to odor. We refer to reviewer to the discussion above (pgs. 1-3) for more details.

I find the paper very well presented, the experiments well described and the results well discussed. In my opinion, however, the main question still remains open: is there really no plasticity in the antennal lobe upon chronic stimulation or is the stimulation not sufficient to drive plasticity? how should the results be reconciled with previous studies? More specific points follow below.1. Did the author try to induce stronger plasticity by using higher concentrations of the odors or continuous delivery? I completely understand the argument for low, natural concentrations, but if one could recapitulate previous results by stronger activation, then these new data could be additively and complementarily interpreted together with previous findings.

We agree with the reviewer’s suggestion that evaluating the impact of exposure to continuous, high concentrations of odor helps with the interpretation of our original results. We now provide this experiment in Figure 4. Our data are most consistent with an interpretation of no or limited plasticity in PNs triggered by chronic stimulation, and, when present, this plasticity is in the opposite direction (mildly enhanced PN responses) to what has been reported previously. At a minimum, chronic odor stimulation does not consistently reduce PN sensitivity for all odors and glomeruli, having provided at least three counter-examples, and the prior model for olfactory plasticity in the antennal lobe is not a general one. Please see the discussion above on pg. 3 for more.

2. Another point I would like to be clarified is whether the PID signal measured in the rearing bottle (fig.1B) is similar to the PID signal measured on the ephys rig. I guess that this is unlikely due to the fact that the PID is not sensitive to the low concentrations used in the ephys.

Some, but not all, of the odor stimuli used on the ephys rig are below the range of sensitivity of the PID. However, when the concentration of the odor source vial is high enough (for instance, 2-butanone at 10^-4^), the odor stimulus can be measured in both setups. In pilot experiments we confirmed that the amplitude of the PID measurement of the odor stimulus is similar on the ephys rig and in the rearing bottle – we did our best to match the olfactometers in the two situations to within ~10%. We do know that the dynamics of the odor stimulus, however, are different between the two setups, because lower flow rates and slightly longer lengths of tubing are used in the odor exposure setup compared to the ephy setup. This is largely for practical reasons, for instance, to avoid drying out the food in the rearing bottle. As a result, the stimulus pulse is more filtered, with slower onset and offset, in the rearing bottle as compared to the snappier, more square shaped pulse on the rig.

Could it be that the low concentrations deplete faster?

I’m not completely sure I understand the principal concern the reviewer has here, but will do my best to provide relevant information. At low concentrations for low vapor pressure odorants, it could take lower to replenish the headspace equilibrium after each stimulus delivery. However, the odors used in this study are all relatively volatile, and the odor stimuli used in the rearing environment are higher in concentration than nearly all odor stimuli used in ephys (since the rearing concentrations elicit near saturating rates of firing). Thus, there is no reason to think odors are depleting faster during rearing than in ephys. Even if some odors at some concentrations in Figure 2 are depleting faster across trial presentations than others, this depletion should be the same in measurements from the two conditions (solvent- versus odorexposed) and shouldn’t account for any systematic differences (or lack there of) that are observed between the two conditions.

Or that the odor stimulus in the rearing bottle does not scale linearly with concentration and therefore no stimulus is actually going through?

The data shown in Figure 1B are direct measurements of the actual odor stimulus (odor and concentration) used for chronic exposure in the bottle exactly as it would be when the flies are placed in the bottle. There is no scaling. The only reason we didn’t do the PID measurement simultaneously with the flies in the bottle during a real experiment is because the flies could potentially damage the sensor. Thus, the measurement in the rearing bottle (Figure 1B) empirically demonstrates that the odor stimulus used for chronic exposure is reaching the interior of the bottle and is relatively stable over many hours. The traces shown in Figure 1B indicate the degree of rundown of the odor stimulus over 24 hrs. We replace the odor source every 12 hours.

3. More in general, I think that it is necessary to demonstrate that the chronic stimulation used in this paper induces some kind of change in sensory perception, for example at a behavioral level. If this is not the case, then one could wonder why to embark in such tedious experiments. If there is instead a behavioral change, then one would logically look for physiological correlates in the AL first and secondly in downstream brain areas.

We agree that demonstrating a behavioral consequence of chronic exposure to low, naturalistic concentrations of monomolecular odor would be great. However, though flies can clearly detect such stimuli at the level of neural activity, they do not behave towards them in isolation, at least in the context of typical lab-based olfactory assays like a trap assay or twochoice assay. There are also many monomolecular stimuli at high concentration to which flies do not show innate behavioral responses in such assays. It does not mean flies cannot perceive these odors, just that they are not motivated to act towards them in these contexts.

We do not agree that a demonstration of behavioral plasticity to our odor exposure protocol is necessary to answer the basic question laid out in the study. Our major goal was to rigorously evaluate whether the dominant neural mechanism proposed to be responsible for olfactory behavioral plasticity – selective decreases in PN sensitivity evoked by chronic excitation – functions in the *Drosophila* antennal lobe. This question is important, because observations of selective PN adaptation to overrepresented odors have been central to the interpretation of behavioral aversion to familiar odors after chronic exposure as behavioral habituation.

In terms of why we would “embark in such tedious experiments,” we believed based on the prior literature in this field that we would observe robust PN adaptation after chronic odor stimulation. We intended to use this tractable experimental model to investigate the cellular and circuit mechanisms mediating glomerulus-specific plasticity; indeed, we expected to be able to elucidate the synaptic and molecular mechanisms by which cell-specific strengthening of GABAergic inhibition could mediate glomerulus/input-specific plasticity. If reduced behavioral aversion to overrepresented odors were really mediated through changes in the odor sensitivity of PNs that process these odors, then strong stimulation of a single ORN input should elicit plasticity selectively in the corresponding PN. Our results suggest that the overall model for odor experience-driven olfactory habituation needs to be revisited, and we refer the reviewer to the Discussion section “Implications for understanding odor-experience dependent changes in olfactory behavior.” The results that we report in this paper are not consistent with the most widely accepted model for how chronic odor exposure affects olfactory behavior. We have reworked the Introduction and Discussion sections to more explicitly discuss these points.

4. The authors end the paper with a discussion about the advantages of not having plasticity. I think that this discussion is a bit biased to justify the current data and does not consider general computational ideas. I would argue that one shouldn't expect plasticity with respect to rather "common" stimuli. The developmental program has evolved to build a sensory system for the "common" environment, plasticity should be a mechanism to cope with variations from the "common". Within this discussion the authors should try clarify what was their initial hypothesis. Why did they expect plasticity in the AL? What did they think this could be useful for?

The reviewer has good points here. To clarify, the section of the Discussion referred to by the reviewer (starting on line 626 in the revised manuscript) focuses on *where* in a sensory circuit it makes sense to have plasticity. We agree that evolutionary pressures generate sensory systems with stimulus selectivities adapted to the “common” environment for the organism. The question addressed by this study is whether, in the case where an organism finds itself in an atypical sensory environment, plasticity that could help the animal adapt to variations from “common” acts primarily at early stages of sensory processing (retina, cochlea, antennal lobe, etc.) or at later stages of sensory processing (visual cortex, auditory cortex, mushroom body, etc.). Our point in this section of the Discussion is that, although there are obvious computational benefits to adapting early stages of sensory processing to the distribution stimuli in the local environment (e.g., the efficient coding hypothesis), there are also some costs, and, on balance, having sensory plasticity become more important at later rather than earlier stages of processing may balance the tradeoffs.

As to the initial hypothesis, we have reworked the Introduction and earlier sections of the Discussion to clarify our expectations and hypotheses. We expected to see plasticity in the AL in large part because multiple past studies had reported plasticity in the AL. This type of plasticity, if present, could be useful as part of a homeostatic process to selectively adjust the sensitivity of the olfactory system to the specific distribution of odors in the local environment. As discussed in the last response and in the Introduction, we intended to use this tractable system for studying the neural mechanisms that would implement glomerulus-specific plasticity in the context of efficient coding on short (within the animal’s lifetime) timescales. However, we unfortunately do not find strong evidence for this form of plasticity in the AL.

5. If the plasticity is broad across PNs, and therefore it is not stimulus specific, how can plasticity depend of valence and context, which require stimulus specific information? (refers to discussion ~L629)

We advance the hypothesis that the weak, broad PN plasticity could be a generalized sensitization of the system in a rich olfactory environment (lines 542-44), which is also seen in other sensory systems in other organisms and is non-stimulus specific. However, we’re suggesting that this weak PN plasticity is not the mechanism that mediates changes in how flies behave towards familiar odors after chronic exposure. Rather, we suggest that the neural mechanism underlying the behavioral plasticity is likely to be downstream of the antennal lobe, for instance, in the mushroom body. The statement about how valence/context might be important for odor experience-dependent plasticity (now lines 622-24) is meant to indicate that additional features of the exposure environment may contribute to how chronic exposure impacts behavior. In other words, the field has largely assumed that chronic odor exposure is a passive process, depending only on sensory activity and not on other information about the environment. There is no reason to assume this is the case, since chronic odor exposure is always done in environments that support the growth of the animal (provision of food, water, often other animals, etc.). These additional features of the environment provide other inputs to the brain that may affect how the animal behaves towards the familiar odor when it is reencountered.

Reviewer #3 (Recommendations for the authors):With electrophysiological and optogenetic experiments performed in *Drosophila*, the authors tested the hypothesis that olfactory codes can adapt to the statistical frequency with which specific odors are encountered in natural environments. Focusing on a small number of glomeruli and their cognate odorants, the authors determined, consistent with prior work, that plasticity in this context is subtle, affects mainly local neurons, with exceptions that reveal no new general plasticity rules. The results suggest that chronic odor exposure increases the overall strength of global excitatory coupling among glomeruli in the antennal lobe after chronic odor exposure, possibly heightening the sensitivity of PNs to weak odors.

We disagree with the reviewer’s characterization that our results were expected and

“consistent with prior work”. Two influential and highly cited studies (Sachse et al. 2007 and Das et al. 2011) predicted we should observe strong reductions in PN responses, which is what we expected to find (see response to Reviewer #2). A more recent study (Kidd et al. 2015) predicted what we actually observed but looked at only one odor at one testing concentration which made determination of the generality of the finding difficult. None of these studies have rigorously tested for the selectivity of PN plasticity, though the selectivity is central to the claim that it mediates experience-dependent behavioral plasticity to familiar odors. We provide a systematic study of three glomeruli with three odors and demonstrate conclusively that PN plasticity acts mildly in the opposite direction from what is commonly accepted, and nonselectively in many PNs (see reviews by Golovin et al. 2016 or Bai and Suziki 2020 for the dominant model in this field). The results require a reconsideration of the most commonly accepted framework for how chronic odor experience changes how flies behave towards familiar odors.

Strengths of this work are that the research question is interesting, the preparation is appropriate and well-used, experiments appear to have been done well, and the authors usefully report what could be described as negative results (more authors should do this!).My main concern is that this work will have limited impact. Similar papers have appeared over the years (many are appropriately cited and described in this manuscript). Each such project featured somewhat different choices of stimuli and protocols, and each arrived at somewhat different conclusions. The authors attribute this variance to "methodological differences" (lines 428ff). The present work contributes a useful new dataset to this field. But it does not directly address or resolve the causes of these different results or yield a new unifying understanding of this form of plasticity.

Please see response to this section repeated below.

With electrophysiological and optogenetic experiments performed in *Drosophila*, the authors tested the hypothesis that olfactory codes can adapt to the statistical frequency with which specific odors are encountered in natural environments. Strengths of this work are that the research question is interesting, the preparation is appropriate and well-used, experiments appear to have been done well, and the authors usefully report what could be described as negative results (more authors should do this!).

We appreciate the reviewer’s summary of the strengths of the study, in particular, that the experiments were executed to a high standard.

My main concern is that this work will have limited impact. A number of similar papers have appeared over the years (many are appropriately cited and described in this manuscript). Each such project featured somewhat different choices of stimuli and protocols, and each arrived at somewhat different conclusions. The authors attribute this to "methodological differences" (lines 428ff). The present work contributes a useful new dataset to this field. But because it does not directly address or resolve the causes of these different results or yield a new unifying understanding of this form of plasticity, I believe only specialists will want to read about it.

We understand the frustration that the reviewer expresses with this field and have felt it ourselves. The large number of odors and glomeruli, and the fact that they lack natural organizational axes, creates significant challenges to developing a systematic understanding of the circuit. The problem is complex, but we can’t just throw up our hands! We believe this study does point towards a new, unifying direction for understanding olfactory behavioral and neural plasticity. In our initial submission, wary of upsetting other labs working in this field whose work we greatly respect, we were not direct enough about our interpretation of our results. This led to a confused and over-nuanced message that was not useful. We still do not think we can conclusively state that prior results are not accurate (Sachse et al. 2007; Das et al. 2011), because we cannot exactly replicate their methods (see above). However, we believe our data firmly supports this conclusion: the commonly accepted idea that chronic excitation reduces PN responses is not true for all glomeruli, and it is probably not true for even most glomeruli. We believe this negative result should be broadly disseminated because it invites a reinterpretation of a large body of behavioral work on how odor experience impacts olfactory behavior.

In the revised manuscript, we are more explicit in the Introduction and Discussion sections about our motivation for the study, the importance of testing the specificity of PN plasticity, and its implications for interpreting the meaning and purpose of odor-experience driven behavioral plasticity. Our basic message is that we don’t believe plasticity in the antennal lobe is the major mechanism mediating odor-experience driven behavioral plasticity, because PN plasticity is rather limited in magnitude. Also, activity-dependent PN adaptation would not correctly account for how animals behave after chronic exposure to attractive odors (see Discussion section, lines 589-605). Our results suggest the need to reinterpret the impact of chronic odor exposure on olfactory behavior as an increase in attraction/acceptance of familiar odors, rather than as habituation. They motivate a search for an alternative neural mechanism to explain behavioral plasticity, likely downstream of antennal lobe processing. The study also highlights the degree of stability of early sensory processing in the insect. This is a problem of general interest to sensory neuroscientists. For instance, a significant body of work in the retina or cochlea examines the limits or bounds of how abnormal sensory experience affects the structure and function of those circuits.

Suggestions for improving the text:1) line 127ff: "Photoionization measurements of the odor stimulus in the rearing bottle demonstrated that the stimulus was stable across more than 24 hours (Figure 1B)." I disagree with this characterization. To me, the PID signals shown in Fig 1B clearly indicates that the amount of released material systematically and substantially decreases over time. The authors can question whether this amount of change is meaningful, but they should not deny the change.

We apologize and agree with the reviewer. We have corrected the sentence as follows: “Photoionization measurements of the odor stimulus in the rearing bottle demonstrated that the stimulus amplitude changed <20% over 12 hours (Figure 1B), after which the odor source was refreshed. The largest decrease in delivered odor concentration occurred within the first two hours and slowed significantly thereafter.” (lines 117-119).

2) line 132: "with little adaptation of the PN response to the odor..." Similarly, Fig 1D also shows systematic decreases over trials and very dramatic adaptation during the course of each response.

We removed this statement.

3) Fig 1C: the authors do not note, but should, that the response duration increases systematically and dramatically over trials.

We added the statement: “PN response duration to the 1 s stimulus increased systematically and substantially across successive odor presentations.” (lines 125-126).

4) line 133ff: "odor stimuli to which we exposed flies were of significantly lower concentration than what has been used in prior studies investigating olfactory plasticity...." This statement calls for more explanation and citations of these prior studies.

We have added the relevant citations and modified the statement to: “Thus, although the odor stimuli to which we exposed flies were of significantly lower concentration than what has been previously used to investigate the effect of chronic odor exposure on PNs, these stimuli drove high, reliable, near saturating levels of firing in their corresponding PN type (Figure 1C, data not shown).”

5) line 177f: "Due to the small size of VM7 PN somata, VM7 PN spikes are small..." I'm not sure I follow this logic: many especially small insect neurons generate especially big spikes, and some big neurons generate small spikelets.

We agree with the reviewer – soma size is probably not the relevant variable. We have removed the clause.

6) line 233f, 405ff, and elsewhere: "This result implicates local lateral circuitry in the antennal lobe in odor-experience dependent neural plasticity." Stopfer and Laurent 1999 (not cited) found similar results in locust exposed to a relatively brief series of odor pulses suggesting many hours of exposure may not be necessary to evoke this plasticity. Subsequent modeling work (Bazhenov et al, 2005, not cited) showed these findings can be explained by activity-dependent facilitation of AL inhibitory synapses, and lead to more reliable odor responses. The authors may wish to consider this work.

We thank the reviewer for pointing us in this direction. We agree that important prior work, including the two interesting studies pointed out by the reviewer, has implicated local AL circuitry in PN plasticity elicited by odor encounters on multiple timescales. We have incorporated consideration of these past results into our Discussion on the mechanisms of plasticity (starting line ~555). It’s indeed interesting to consider whether repeated odor encounters on relatively short timescales are sufficient to evoke the PN plasticity we see. In addition to Stopfer and Laurent, 1999 and Bazhenov et al. 2005 that asked this question in the locust AL (which has robust LFP oscillations, a feature not present in the Drosophial AL), a recent study by Franco and Yaksi, 2022 examines the question explicitly in *Drosophila* PNs. Like in locust, they observe that repeated odor stimulation (over seconds to minutes) leads to adaptation of PN response amplitude and more reliable odor representations. This short-term plasticity in the *Drosophila* antennal lobe, however, appears to act in the opposite direction to long term plasticity in terms of the impact on PN response strength elicited by chronic odor exposure.

7) lines 259f: "Unexpectedly, in flies chronically exposed to E2-hexenal only, many glomeruli were smaller in volume than their counterparts in solvent-exposed flies." This is a potentially worrisome indication that the flies may have been adversely affected in a general way by the chronic exposure. Did the exposed flies appear healthy compared to controls? Can the authors rule out that they had become sick?

Flies chronically exposed to E2-hexenal (10^-7^, e.g., 0.00001%) appear qualitatively normal – they walk, climb the sides of the vial, and attempt to escape in a manner indistinguishable from unexposed flies. The experimentalist masked to condition cannot distinguish between flies from the exposure groups. In contrast, in our hands, exposure to some odors at high concentration (e.g., 20% ethyl butyrate) can cause flies to appear lethargic, unhealthy, and reduce their locomotion (see above).